# Mesoscale fine structure of a tropopause fold over mountains

Wolfgang Woiwode[1], Andreas Dörnbrack[2], Martina Bramberger[2], Felix Friedl-Vallon[1], Florian Haenel[1], Michael Höpfner[1], Sören Johansson[1], Erik Kretschmer[1], Isabell Krisch[3], Thomas Latzko[1], Hermann Oelhaf[1], Johannes Orphal[1], Peter Preusse[3], Björn-Martin Sinnhuber[1], and Jörn Ungermann[3]

[1]Institute of Meteorology and Climate Research, Karlsruhe Institute of Technology, Karlsruhe, Germany
[2]Deutsches Zentrum für Luft- und Raumfahrt, Institut für Physik der Atmosphäre, Oberpfaffenhofen, Germany
[3]Forschungszentrum Jülich, Institute of Energy- and Climate Research, Stratosphere (IEK-7), Jülich, Germany

*Correspondence to*: Wolfgang Woiwode (wolfgang.woiwode@kit.edu)

**Abstract.** We report airborne remote-sensing observations of a tropopause fold during two crossings of the polar front jet over Northern Italy on 12 January 2016. The GLORIA (Gimballed Limb Observer for Radiance Imaging of the Atmosphere) observations allowed for a simultaneous mapping of temperature, water vapour and ozone. They revealed deep, dry and ozone rich intrusions into the troposphere. The mesoscale fine structures of dry filaments at the cyclonic shear side north of the jet and tongues of moist air entraining tropospheric air into the stratosphere along the anticyclonic shear side south of the jet were clearly resolved by GLORIA observations. Vertically propagating mountain waves with recorded temperature residuals exceeding $\pm$ 3 K were detected above the Apennines. Their presence enhanced gradients of all variables locally in the vicinity of the tropopause. The combination of $H_2O$-$O_3$-correlations with potential temperature reveals an active mixing region and shows clear evidence of troposphere-to-stratosphere and stratosphere-to-troposphere exchange. High-resolution short-term deterministic forecasts of ECMWF's integrated forecast system (IFS) applying GLORIA's observational filter reproduce location, shape, and depth of the tropopause fold very well. The fine structure of the mixing region, however, cannot be reproduced even with the used 9 km horizontal resolution of the IFS. This case study demonstrates convincingly the capabilities of linear limb-imaging observations to resolve mesoscale fine structures in the upper troposphere and lower stratosphere, validates the high quality of the IFS data, and suggests that mountain wave perturbations have the potential to modulate exchange processes in the vicinity of tropopause folds.

## 1 Introduction

Tropopause folds are preferred regions of bidirectional stratosphere-troposphere exchange (STE) of mass and trace gases in the middle latitudes (e.g., Holton et al., 1995, Gettelman et al., 2011, and references therein). Their generation is related to cyclogenetically active regions (e.g. Keyser and Shapiro, 1986 and references therein) which develop narrow transition zones where all flow variables become very concentrated. The associated formation of upper-level fronts and surface fronts is mainly driven by the nonlinear self-advection of evolving Rossby waves. Tropopause folds form preferentially along strong jet streams separating polar or Arctic air masses from those of subtropical origin. The polar front jet (PFJ) is found at middle latitudes and is often classified as eddy-driven jet stream (Held, 1975, Lee and Kim, 2003). Upper-level jet/front systems are prone to the generation of clear-air turbulence which is an important exchange mechanism for atmospheric constituents across the tropopause (e.g., Shapiro, 1978, 1980, Koch et al., 2005, Kühnlein, 2006, Sharman et al., 2012). Extratropical jet streams are waveguides for planetary Rossby waves (e.g., Dritschel and McIntyre, 2008, Martius et al., 2010). Moreover, they provide a favourable medium for the vertical propagation of internal gravity waves excited in the troposphere (e.g., Preusse et al. 2006, Ern et al. 2018).

Observations of tropopause folds go back to the middle of the last century when their spatial structure was portrayed by observations from radio soundings distributed over large horizontal distances (Reed, 1955, Reed and Danielsen, 1958, historical review by Keyser and Shapiro, 1986). Later, and due to the urgent need to explain the exceptional maximum springtime radioactive fallout over North America, the first coordinated aircraft observations were undertaken (Danielsen 1964, 1968, Danielsen and Mohnen, 1977). Besides atmospheric variables as wind and temperature, they sampled trace gases, radio nuclei, and aerosols along stacked flight legs through jet streams. Their composites combining potential temperature, potential vorticity, and various constituents enhanced our knowledge about the spatial structure of tropopause folds and the contributions of advection, mixing, and radiation to STE. Melvin Shapiro brought those early conceptual plots by Reed and Danielsen to perfection and designed 2D cross-sections of potential temperature, wind, ozone and condensation nuclei from flight level and dropsonde measurements (Shapiro, 1980). Basically, these conceptual visualisations provided the base for modified and refined schematics in recent publications (e.g., see Fig. 1 in Gettelmann et al., 2011). Later, airborne lidar observations using the Differential Absorption Lidar technique provided valuable insights into the spatial structure of tropopause folds (e.g. Browell, 1987, Ehret, 1999, Flentje, 2005). Airborne observations by one lidar mostly give mixing ratios of one atmospheric constituent (e.g., water vapour or ozone) and aerosol backscatter. Simultaneous airborne lidar observations of ozone and water vapour were only possible

on large platforms as NASA's DC8 (Kooi et al., 2008), and, most recently, on the German research aircraft HALO during a mission over the North Atlantic (A. Schäfler, pers. comm., 2018).

Limb observations of various trace gases from a high-flying research aircraft were analysed by Weigel et al. (2012) and Ungermann et al. (2013) to investigate their filamentary structure in the summer-time extratropical transition

layer near the subtropical jet stream (STJ). The Cryogenic Infrared Spectrometers and Telescope for the Atmosphere – New Frontiers (CRISTA-NF) resolved filaments of a spatial scale of less than 800 m vertically. The diagnostics applied (tracer-tracer correlations of PAN and $O_3$) provided the first multi-species 2D portrait of the inhomogeneous distributions within a tropopause fold extending from about 14 km down to 8 km altitude. Moreover, infrared limb observations are capable of resolving gravity waves in the temperature field (e.g., Preusse

et al., 2002; Ungermann et al., 2010). Airborne temperature measurements by the Gimballed Limb Observer for Radiance Imaging of the Atmosphere (GLORIA) in the tomographic mode (Ungermann et al., 2011) were recently used to characterize vertically propagating mountain waves above Iceland (Krisch et al., 2017). GLORIA is a limb-imaging Fourier transform spectrometer for atmospheric research in the upper troposphere and lower stratosphere (UTLS) which was developed as a precursor for future satellite missions (Friedl-Vallon et al., 2014,

Riese et al., 2014). Compared to conventional limb scanning instruments (e.g. Fischer et al., 2008), GLORIA's limb-imaging technique enables much higher sampling rates as all vertical viewing angles associated with a set of atmospheric parameter profiles are measured at the same time.

The horizontal viewing characteristic of GLORIA plays an important role. In cases of horizontally elongated features (i.e. several hundreds of kilometres) such as tropopause folds along the jet stream, the along-track

sampling of GLORIA can resolve vertical cross-sections with high detail by choosing  flight tracks perpendicular to the jet axis. Thereby, GLORIA's low horizontal direction resolution along the line-of-sight is directed along the elongated atmospheric feature, while the dense along-track sampling is exploited to resolve structures across the jet axis.

In the past, temperature in tropopause folds along the vertical direction was observed at flight level and by

dropsondes underneath. Simultaneous in situ measurements of both trace gases and temperature have not been applied up to now to characterize the mesoscale fine structure of tropopause folds. CRISTA-NF observations in July 2006 (Weigel et al., 2012) provided a first coarse perspective of a tropopause fold using this combination of parameters. Here, we present GLORIA airborne observations of temperature, water vapour, and ozone taken simultaneously during two passages of a tropopause fold located over Northern Italy on 12 January 2016 in

unprecedented quality. GLORIA was deployed on board the High Altitude and Long Range Research Aircraft

(HALO) during the merged campaigns POLSTRACC (POLar STRAtosphere in a Changing Climate), GW-LCYCLE II (Investigation of the life cycle of gravity waves), and SALSA (Seasonality of Air mass transport and origin in the Lowermost Stratosphere using the HALO Aircraft), in the following abbreviated as PGS. Goals of the PGS campaign were atmospheric and chemical observations related to the Arctic stratospheric polar vortex, gravity waves, and the seasonality of air mass exchange in the UTLS.

GLORIA observations along two extended legs of the research flight over Northern Italy reveal the existence of a deep stratospheric intrusion wrapping around the PFJ. At the same time, mountain waves excited by the strong low-level flow over the Apennines propagated vertically from the troposphere into the stratosphere. Both, the south- and northbound flight legs were oriented nearly perpendicularly to the axis of the PFJ. Therefore, GLORIA probed the tropopause fold with a propitious viewing geometry pointing nearly parallel into the elongated, zonally oriented jet stream. Low water vapour and high ozone volume mixing ratios were observed inside the tropopause fold. Additionally, mountain-wave induced temperature anomalies in the vicinity of the PFJ and the tropopause fold could be captured.

Admittedly, observations of interactions of jet streams and, in particular, tropopause folds with gravity waves are sparse (e.g. Buss et al., 2004, Koch et al. 2005). Thus, GLORIA's airborne observations presented in this paper have the potential to enhance our knowledge about tropopause folds interfering with vertically propagating mountain waves. With a view on exchange processes between stratosphere and troposphere, observations of gravity waves interfering with jet streams and developed tropopause folds are of particular interest. Especially in cases of widespread mountain wave excitation above large mountain ranges, mountain wave-jet stream interactions may significantly affect mixing processes associated with tropopause folds. Most recently, Heller et al. (2017) combined different methods to quantify the location, direction and irreversibility of the water vapour transport during a strong mountain wave event during the Deep Propagating Gravity Wave Experiment, Fritts et al. (2016). Both large positive and negative vertical water vapour fluxes were detected at flight level above and in the lee of the Southern Alps of New Zealand. Tracer-tracer correlations of water vapour to ozone were used to indicate the vertical transport followed by irreversible mixing of water vapour. Their analysis was based on in-situ measurements providing tracer-tracer correlations in dependence of potential temperature at flight levels; see their Figure 8. Here, GLORIA observations offer the possibility to construct tracer-tracer correlations and to present them as function of potential temperature at different altitudes below the flight path for the first time.

Nowadays, numerical weather prediction (NWP) models such as the Integrated Forecast System (IFS) of the ECMWF are capable of simulating tropospheric and stratospheric dynamics including tropopause folding and

gravity waves with high spatial and temporal resolution (e.g. Preusse et al., 2014, Dörnbrack et al., 2017, references therein). Detailed studies using observations from particular field campaigns or from operational ground-based or satellite sensors are required to validate gravity wave parameterizations of numerical weather forecast systems and models for climate projection (Fritts and Alexander, 2003; Alexander et al. 2010; Geller et al. 2013; Fritts et al., 2016). As we use linear limb observations (i.e. viewing perpendicular to the flight path, without azimuth panning), the IFS data may be interpolated directly at the tangent points. On the other hand, in case of confined local features (i.e. several tens to a few hundreds of kilometres) such as mountain wave-induced temperature modulations, GLORIA's observational filter, i.e. its smoothing characteristics in the domain along viewing direction must be characterized and applied to the IFS data for a meaningful comparison (Ungermann et al., 2011). Here, the IFS data are folded with GLORIA-specific observational filters for water vapour ($H_2O$) and temperature (T) for quantitative comparisons with the GLORIA data.

Altogether, combining GLORIA data, 1-hourly short-term deterministic IFS forecasts, and in-situ observations we analyse how realistically the IFS reproduces the observations. Furthermore, we discuss the tropopause fold-mountain wave interference and implications with respect to stratosphere-troposphere exchange. The paper is structured around the following research questions:

- Do the GLORIA observations of water vapour, ozone, temperature and potential temperature follow the accepted conceptual models of tropopause folds? How do mountain waves modify the temperature and trace gas distribution?
- Can active mixing regions and STE be identified by means of the infrared limb sounding? Where are they located with respect to the tropopause fold?
- How do high-resolution IFS data compare with the observations? Do observational filters from GLORIA applied to the IFS data improve the comparison with the measurements?

Section 2 describes the data and the methodology to analyse them. Section 3 summarizes the meteorological situation under which the observations took place. Section 4 presents GLORIA data and compares the observations in the vicinity of the aircraft with the flight level in-situ data and selected vertical profiles with one available radiosonde sounding. The core results of the paper are presented in Section 5 which considers the mesoscale fine

structure of the observed tropopause fold, the mountain-wave induced temperature fluctuations, and the mixing across the tropopause. Section 6 concludes the paper.

## 2 Methodology

### 2.1 GLORIA observations

GLORIA is a cryogenic limb-imaging spectrometer deployed on board high-altitude aircraft (Friedl-Vallon et al., 2014; Riese et al., 2014). Measurements are performed in limb mode (Fig. 1) to the right hand side of the flight track. Using 128 vertical × 48 horizontal pixels of a HgCdTe detector, GLORIA passively observes the thermal radiation of the atmosphere in the spectral range from 780 $cm^{-1}$ to 1400 $cm^{-1}$. The pointing of GLORIA is stabilized using a gimballed frame, aided by an inertial navigation system. During each interferometer sweep, GLORIA

records data cubes of interferograms with all pixels simultaneously. Thereby, the detector rows correspond with limb-viewing geometries with tangent altitudes between typically ~5 km and flight level. In post-flight data processing, spectra of the detector rows of each data cube are binned to reduce uncertainties. The spectra are quantitatively calibrated using in-flight blackbody measurements (Kleinert et al., 2014).

GLORIA can be operated with different spectral sampling rates. Thereby, a higher spectral sampling results in a

lower along-track sampling. Here, we use measurements in high-spectral resolution mode ("chemistry mode") with a spectral sampling of 0.0625 $cm^{-1}$. The resulting apodized spectral resolution of 0.121 $cm^{-1}$ (full width at half maximum) is particularly useful for resolving weak and narrow spectral signatures of minor species. In this measurement mode, one data cube resulting in one vertical sequence of calibrated spectra is recorded within 13 s. This corresponds with a net along-track sampling of ~3 km.

The retrieval of atmospheric parameters from the high-spectral resolution mode observations during PGS and their validation are reported by Johansson et al. (2018). Prior to the retrieval, the binned spectra are cloud-filtered according to the cloud index method by Spang et al. (2004). A variable threshold value is applied, increasing linearly from 3.0 for the lowest limb-views to 1.8 at flight altitude. For the retrievals, the radiative transfer model KOPRA (Karlsruhe Optimized and Precise Radiation transfer Algorithm; Stiller et al., 2002) is used in

combination with the inversion algorithm KOPRAFIT (Höpfner et al., 2001). Temperature is retrieved using two times two rotational-vibrational transitions of $CO_2$ in the microwindows from 810.5 $cm^{-1}$ to 812.9 $cm^{-1}$ and 956.0 $cm^{-1}$ to 958.2 $cm^{-1}$. These microwindows show a sufficient transparency at low altitudes. The used spectral transitions are suited well for a temperature retrieval, since they are sufficiently strong, clearly separable from other signatures, and characterized by different opacities and different temperature dependences. In the

temperature retrieval, IFS high-resolution (HRES) operational analyses spectrally truncated at wavenumber 213 and at 137 vertical hybrid levels are used as initial guess and a priori information. The operational ECMWF data was interpolated directly to the GLORIA tangent points. The data are available every six hours and are hereafter referred to as "HRES". All retrievals are based on geometric altitude levels. Associated pressures are interpolated

from the HRES data. GLORIA's potential temperature is calculated from the retrieved temperature and the HRES pressure field.

Ozone volume mixing ratio is retrieved using several rotational-vibrational transitions in the microwindows from 780.6 cm$^{-1}$ to 781.7 cm$^{-1}$ and 787.0 cm$^{-1}$ to 787.6 cm$^{-1}$. The ozone initial guess and a priori profile is taken from Remedios et al. (2007). The natural logarithm of water vapour volume mixing ratio is retrieved using a single

rotational transition in the microwindow from 795.7 cm$^{-1}$ to 796.1 cm$^{-1}$ and then converted to volume mixing ratio. For initial guess and a priori a vertically constant profile is applied; the value is the logarithm of 10.0 ppmv. In all cases, narrow microwindows are chosen to minimize spectral interference with pressure-broadened signatures of other gases at lower altitudes. Typical vertical resolutions between 300 m and 900 m are achieved between flight altitude and the lowest tangent point. Finally, the retrieved profiles are combined to 2-dimensional vertical cross

sections of the respective target parameters along the flight track.

**2.2 ECMWF IFS forecasts**

The IFS model is a global, hydrostatic semi-implicit, semi-Lagrangean NWP model. The IFS cycle 41r1 was operational from 12 May 2015 until 8 March 2016 and utilized a linear grid with a spectral truncation at wavenumber 1279 (T$_L$1279) which corresponds to a horizontal resolution of approximately 16 km. In the vertical,

137 levels (L137) ranged from the model top at a pressure level of 0.01 hPa down to the surface. The vertical resolution in the vicinity of the extra-tropical tropopause is less than 400 m and increases with decreasing altitude. The IFS cycle 41r1 was replaced by cycle 41r2 on 8 March 2016. The horizontal resolution of all the different operational applications using the IFS were upgraded (Hólm et al., 2016). The deterministic high-resolution analyses and forecasts are computed on a cubic octahedral grid with a resolution of approximately 9 km while the

spectral truncation remained at wavenumber 1279 (T$_L$1279), Malardel and Wedi (2016). A large contribution to the gain in effective resolution of the IFS cycle 41r2 results from the reduced numerical filtering and the preparation of the physiographic data at the surface.

IFS cycle 41r2 was running in a pre-operational, experimental suite in January 2016. Here, one hourly short-term forecasts of the 00 UTC run at 09 UTC, 10 UTC, and 11 UTC are used. These data were interpolated to a regular

0.25° × 0.25° latitude-longitude grid and will be compared to the GLORIA observations. The IFS data are retrieved

as fully resolved fields, containing all 1279 spectral coefficients. Background temperature fields are retrieved as moving average within a horizontal window of 3° in meridional direction and 4° in zonal direction (~330 km x 330 km). The difference of the fully resolved IFS temperature field (or the GLORIA temperature field) and the IFS background temperature field yields perturbations which accentuate mesoscale temperature variations, e.g. due to mountain waves.

Studies using the same IFS cycle are reported in the literature: Ehard et al. (2018) documented the high fidelity of the IFS fields representing the mean stratospheric temperature over Sodankylä, Finnland (67.5°N, 26.5°E) in December 2015 and the gravity activity at this location in the winter 2015/2016. Dörnbrack et al. (2017) compared the mesoscale structure of mountain-wave induced polar stratospheric clouds with the simulated temperatures of the IFS and found a remarkable agreement with the space-borne measurements.

**2.3 GLORIA observational filters**

There are two ways to compare the GLORIA data resulting from radiance integrated along extended limb views (Fig. 1) with the numerical IFS fields. First, the time-dependent 3D IFS data are linearly interpolated in space and time to the GLORIA tangent points which are provided as function of latitude, longitude, altitude, and time. This procedure does not take into account the horizontal smoothing characteristics intrinsic to the GLORIA observations. As a second approach, the observational filters of the GLORIA observations are characterized. In effect, we calculate exemplary 2D averaging kernels (Ungermann et al, 2011) providing the horizontal and vertical smoothing characteristic along GLORIA's viewing direction (i.e. perpendicular to the flight track). These averaging kernels are then used to smooth the IFS data for comparison. Whereas the first approach is straightforward, the second one requires the following calculation of the 2D averaging kernels.

According to Rodgers (2000), the averaging kernel matrix $A \in \mathbb{R}^{n \times n}$ is the product of the gain matrix $G \in \mathbb{R}^{n \times m}$ and the Jacobi matrix $K \in \mathbb{R}^{m \times n}$. Thereby $n$ is the number of retrieval grid levels (i.e. altitudes) and $m$ the number of elements of the measurement vector (i.e. spectral grid points within the microwindows used, for all vertical viewing angles of the measurement). By multiplying $A$ with the delta between an estimate of the true atmospheric state $x$ (e.g. model data or in situ profile) and the a priori profile $x_a$ used for the retrieval, the atmospheric state projected by the retrieval $\hat{x}$ (i.e. smoothed model or in situ profile) can be calculated as follows (measurement errors neglected):

$$\hat{x} = x_a + A(x - x_a)$$

(1)

This approach is often used for comparing in situ observations or model data of high vertical resolution with a retrieval result with a lower vertical resolution. However, this approach involves a single profile representing the true atmospheric state and assumes homogenous conditions along the viewing direction.

According to Ungermann et al. (2011), a number of profiles representing the true state along the viewing direction can be involved in cases where this condition is not fulfilled. A modified averaging kernel matrix $\widetilde{A} \in \mathbb{R}^{n \times (n \cdot o)}$ can be used to replace $A$ in equation (1), with $o$ representing the number of profiles used to sample the variation along the additional dimension along the viewing direction. The 2D averaging kernel matrix $\widetilde{A}$ is calculated as the product of $G$ and a modified Jacobi matrix $\widetilde{K} \in \mathbb{R}^{m \times (n \cdot o)}$ including the additional horizontal dimension. The individual Jacobi matrix elements are calculated for all elements of the measurement vector and with reference to all discretized retrieval grid levels (vertical domain) and locations along the viewing direction (horizontal domain). We calculate $\widetilde{K}$ using the 3D inhomogeneous radiative transfer routine of KOPRA (von Clarmann et al., 2009). The calculations of the elements of $\widetilde{K}$ are extensive, since the retrieval microwindows used in the GLORIA high spectral resolution mode include many spectral grid points. Together with the large number of viewing angles of a single GLORIA data cube (i.e. up to 128, depending on quality and cloud filtering) large sizes result for $\widetilde{K}$.

Our aim is to apply the horizontal smoothing procedure for water vapour and temperature and only for two subsections of the flight (i.e. the two passages of the tropopause fold). Therefore, we calculate $\widetilde{K}$ and $\widetilde{A}$ for each target parameter only for a single observation characteristic for the respective flight passage. The resulting observational filters $\widetilde{A}$ for water vapour and temperature, respectively, are then applied to sample the IFS data of the entire passage. A horizontal discretization of 25 km is chosen to calculate the elements of $\widetilde{K}$ along the horizontal dimension, i.e. along the viewing direction. Since the calculation of $\widetilde{K}$ takes into account the entire light paths of the limb views up to the top of the atmosphere, the discretized elements of $\widetilde{K}$ are calculated within 1500 km along viewing direction. The resulting contributions of $\widetilde{A}$ far beyond the tangent points are however negligible. Therefore, elements of $\widetilde{A}$ beyond 600 km are omitted in IFS data sampling.

In case of temperature, $\widetilde{A}$ is used together with $x$ (including the $o$ model profiles along viewing direction) and $x_a$ (including $o$ times the a priori) to calculate a single smoothed IFS profile $\hat{x}$ for the comparison with a single corresponding GLORIA profile. In case of water vapour, $\widetilde{A}$ is calculated with respect to the natural logarithm of the volume mixing ratio. Since the 3D inhomogeneous radiative transfer routine of KOPRA provides the Jacobi matrix elements corresponding with volume mixing ratio and not the logarithm, the matrix elements of $\widetilde{K}$ are post-

differentiated with respect to the logarithm of volume mixing ratio. The logarithms of $x$ and $x_a$ are then used in the smoothing procedure of the IFS data.

Figure 2 shows exemplary rows of $\widetilde{A}$ for water vapour and temperature for the first tropopause fold passage (figuratively: $n$ times $o$ horizontal profiles of a single row of $\widetilde{A}$, stacked above each other). The individual panels show how a single state element of the retrieval result (i.e. target parameter at altitude level indicated on the top of a panel) responds to an atmospheric grid point along viewing direction (characterized by geometric altitude and horizontal distance) in the true state (Ungermann, 2011; Ungermann et al., 2011). Thereby, "constructive" contributions (red) correspond with atmospheric grid points where a higher/lower value in the true state results in a higher/lower value in the retrieval result. On the other hand, "destructive" contributions (blue) correspond with atmospheric grid points where a higher/lower value in the true state, anti-intuitively, results in lower/higher value in the retrieval result. In this manner, the response of the retrieval and its weighting functions in the vertical and horizontal domain along the viewing direction are characterised.

In Figure 2, $\widetilde{A}$ is calculated for the measurement at a geolocation at 725 km along flight path (compare Fig. 8). Thereby, the latitude of the uppermost tangent points is ~42.46°N (compare Fig. 6). For the second passage (not shown), $\widetilde{A}$ was calculated for the measurement at a geolocation at 2852 km along flight path, with the uppermost tangent points at ~42.33°N. In case of water vapour (Figs. 2a, c, and e), the plots show the response of the logarithm of water vapour at the indicated retrieval grid level to variations of the logarithm of water vapour in the true state (i.e. interpolated IFS data) along the viewing direction. In case of temperature, the response of retrieved temperature at the indicated level to variations of temperature in the true state along viewing direction is shown in the same manner (Figs. 2b, d, and f).

Figures 2a and 2b show the situation approximately at a retrieval grid level of 12.25 km, which approximately coincides with the flight altitude (~12.40 km) during the measurement. In case of the logarithm of water vapour (Fig. 2a), the retrieval result is dominated by constructive contributions within an arched lobe peaking ~50-100 km away from the observer. Further significant constructive contributions originate from locations closer to the observer position, and also a long tailing towards higher altitudes further away is noted. Another significant destructive lobe is found above the dominating constructive lobe. Here, the presence of more water vapour in the true state diminishes the retrieval result at the indicated level. For temperature (Fig. 2b), the result is similar to the logarithm of water vapour. Here, the constructive lobe peaks slightly closer to the observer position at a distance of ~25-75 km.

The response at the retrieval grid level of 10 km is shown in Figures 2c and 2d. For both the logarithm of water vapour, and temperature, the arched constructive lobes are centred approximately at the corresponding retrieval grid tangent point. However, the main maxima are shifted by about 50 km towards the observer and a few hundreds of meters to higher altitudes. Behind the tangent point, the constructive lobes extend further to higher altitudes in both cases. Furthermore, a weak destructive lobe is found on top of the dominating constructive lobe in both cases. Overall, the bulk response is found in a region roughly within about ±100 km around the tangent point (i.e. shifted slightly to the observer in the case of temperature).

The responses at 8 km shown in Figures 2e and 2f show a similar pattern. In case of the logarithm of water vapour (Fig. 2e), again a constructive main lobe and a weak destructive lobe are found. Their maxima are aligned more symmetrically around the corresponding tangent point when compared to Figures 2c and 2d. In case of temperature (Fig. 2f), both, the constructive main lobe and the weaker destructive lobe are developed stronger at the observer-facing side of the tangent point, with the extrema shifted by ~50 km to the observer. Again, the bulk response originates from a region within about ±100 km around the tangent point (i.e. in the case of temperature shifted slightly to the observer).

Overall, the results in Figure 2 show that the retrieval results at certain altitudes are affected significantly by regions around and above the tangent points as a consequence of the viewing geometry and the retrieval algorithm. This is of particular importance for comparisons with highly resolved model data including local fine structures (see Fig. 6b). In the following, we use exemplary observational filters $\tilde{A}$ both, for water vapour and temperature, calculated for the respective tropopause fold passages for detailed comparisons with the IFS.

## 2.4 In-situ observations

The temperature data retrieved from the GLORIA observations is compared with static air temperature at flight altitude provided by HALO's Basic HALO Measurement and Sensor System (BAHAMAS, Krautstrunk and Giez, 2012). The static air temperature data is characterized by a total uncertainty of 0.5 K and is available with a temporal resolution of 1 s. Furthermore, vertical profiles of temperature and water vapour retrieved from the GLORIA measurements are compared with the radiosonde profile of the 12Z (11 UTC) launch from LIRE Pratica Di Mare (station number 16245) at the flight day (available at http://weather.uwyo.edu/upperair/sounding.html). Typical radiosonde measurement uncertainties are 0.4 to 1 K for temperature, around 24 % for water vapour below -50°C and between 5 % and 14 % at higher temperatures (Nash, 2015). However, low stratospheric water vapour mixing ratios are not resolved by the radiosonde data.

**3 Meteorological conditions during research flight PGS06**

Research flight PGS06 was designated as a survey and ferry flight from Oberpfaffenhofen, Germany (48°N, 11°E) via Malta (36°N,14°E), and, eventually, to Kiruna, Sweden (68°N, 20°E). HALO took off at 07:56 UTC and landed at 16:49 UTC on 12 January 2016. The total flight track along with GLORIA's tangent points is shown in Figure 3a. In this paper, we focus on the two legs A-B and C-D. On the southbound leg A-B, atmospheric observations by GLORIA pointed westwards and started above Northern Italy at ~08:25 UTC reaching way point B at 09:52 UTC. On the northbound transect C-D, GLORIA pointed east towards the preceding leg A-B and took observations from 10:26 UTC to 11:54 UTC.

Figure 3b displays the vertical profile of flight track projected onto 13°E along the wide-ranging latitude band flown during PGS06. To set this research flight into a meteorological context, lines of constant potential temperature Θ (isentropes), the height of the dynamical tropopause (2 PVU line), and selected contour lines of constant wind are added from the IFS data valid at 12 January 2016 10 UTC. The atmospheric flow consists of two dominating jets. Between 50°N and 60°N, the horizontal wind of the stratospheric polar night jet (PNJ) attains more than 120 m s$^{-1}$ at about 40 km altitude (not shown). At its lower and equatorward edge, the PNJ merges with the polar front jet (PFJ) which is south of the Alps at about 42° N (Fig. 3b). Here, horizontal winds maximize with about 70 m s$^{-1}$ at 10 km altitude. These jets mark roughly the edge of the Arctic polar vortex. At the jet axis, there is a sharp jump in the height of the dynamical tropopause. On isentropic surfaces, say at 320 K, this discontinuity in PV between the cyclonic (northward) and anticyclonic (southward) shear restricts the transitions of air masses between the stratosphere (north) and troposphere (south). Generally, intrusions of stratospheric air into the troposphere proceed in stably stratified layers with high PV values (Shapiro et al., 1980). As the tropopause wraps or folds around the jet core, such patterns are usually called tropopause folds. Underneath the core of the PFJ, the strongly tilted isentropic surfaces mark the associated baroclinic frontal zone. Except for the appearance of stratospheric gravity waves above and poleward of the core of PFJ, the cross-section as presented in Fig. 3b resembles the schematic sketch by Shapiro et al. (1987, Fig. 17) of the locations and altitudes of frontal systems. Even the Arctic front near 70° N is visible in Fig. 3b.

At the beginning of the southbound leg A-B, HALO climbed up to the lowermost stratosphere and reached the Θ ≈ 360 K isentropic surface at the cyclonic side of PFJ. After overflying the tropopause fold as indicated by the convoluted 2 PVU contour line underneath the flight level, HALO intersected the 350 K isentropic surface at the anticyclonic side of the PFJ. During the northbound leg C-D, HALO reached again the 360 K isentropic surface and climbed up to higher altitudes. Isentropic levels of Θ ≈ 370 K and 380 K were reached further north at 44°N

and 47°N, respectively, as a consequence of the lower troposphere. During the long transect toward way point E in the Arctic, HALO further climbed to higher altitudes and at the same time entered again the 360 K isentrope due to the overall meridional gradient of the isentropes.

As outlined by Bramberger et al. (2018, their Figure 3), the synoptic situation was characterized by a large-scale upper-level trough above Northern Italy and mid-Europe leading to a remarkable North-South gradient of the height of the dynamical tropopause along the flight track (Fig. 3b). Here, Figure 4 juxtaposes horizontal cross sections of water vapour and horizontal wind at the 320 hPa pressure surface (≈ 9 km altitude) of the IFS forecasts valid at 09 UTC and 11 UTC, respectively. As expected, there is a large meridional gradient in water vapour with lower values toward the north and higher values in the south (Fig. 4a, b). At the cyclonic shear side of the PFJ, the lower water vapour mixing ratios appear more textured most probably due to frontal convection and the interaction of the weather system with the Alps. South of the Alps and north of about 43°N, a zonally elongated, nearly homogeneous band of very low water vapour values less than 10 ppmv (deep blue colours in Figs. 4a, b) indicates the dry intrusion of stratospheric air associated with the tropopause fold. The PFJ and the associated tropopause fold extended zonally between 44°N and 42°N from Southern France to Italy (Fig. 4c, d). Maximum horizontal winds of the PFJ exceeded 70 m s$^{-1}$ at this level and are located south of this dry intrusion. During the 3.5 hours when research flight PGS06 travelled between way points A and D, the whole structure of the wind and water vapour fields changed only marginally as the meteorological system propagated slowly south-eastward. Therefore, we can assume that GLORIA sampled air masses in the same meteorological situation on the south- and northbound legs, respectively. Possible differences in the observations must be associated with local atmospheric processes along and underneath the respective observational paths. Additionally, as obvious from the orientations of the individual flight legs, the viewing directions are not parallel in both legs and cut the tropopause fold differently, compare Fig. 3a and Fig. 4.

Figure 5 displays the vertical wind and the temperature perturbations at the 180 hPa pressure surface (≈ 13 km altitude) which corresponds approximately to HALO's flight level on 12 January 2016. The strong westerly low-level flow excited mountain waves over the Pyrenees, the French Alps, Corsica/Sardinia and the Apennines (Bramberger et al., 2018). The corresponding up- and downdrafts remained nearly stationary in the period considered but their amplitude attenuated slightly in time (Fig. 5a, b). Similar but considerably broader patterns are also identified in the mountain-wave induced temperature perturbations (Fig. 5c, d). As in the case of vertical wind, the major structures remained mostly stationary, while amplitudes and patterns changed on smaller scales. The amplitude of the temperature perturbation faded somewhat out at 11 UTC (Fig. 5d). A positive temperature

perturbation is enframed by HALO's south- and northbound legs between 42°N and 43°N. This temperature anomaly is located above the core of the jet stream (cf. Fig. 4a) and is well covered by the GLORIA observations along the southbound flight leg A-B. As explained in detail by Bramberger et al. (2018), mountain-wave induced flow perturbations as represented by the local down- and updrafts above the Apennines are responsible for the observed changes of ambient temperature along the southbound flight leg.

Figure 6 presents longitude-altitude sections of water vapour, horizontal and vertical winds, and temperature perturbations of the IFS data at 42.46°N and valid at 09 UTC when the mountain wave amplitudes were maximum in the IFS (Fig. 5c). Also shown is a projection of GLORIA's tangent points (data cube with uppermost tangent point coinciding with the indicated latitude). As mentioned above, GLORIA's viewing direction was not exactly east-west but approximately towards south-west during the southbound leg. The water vapour distribution reveals the existence of a dry stratospheric intrusion associated with the tropopause fold at about 400 hPa and west of about 12°E. At higher levels, the dry stratospheric air stretched across the whole Italian peninsula (Fig. 6a). Thus, GLORIA viewed along a zonally rather homogeneous water vapour distribution at this latitude above 280 hPa. As GLORIA viewed nearly perpendicular to the strong North-South water vapour gradient (see Fig. 4), the successive measurements along the flight track sampled the meridional structure densely and, eventually, resolved the meridional variation inside the tropopause fold very sharply. At the same time, the blurred horizontal focus along viewing direction was directed nearly in zonal direction into a much smaller variability, leading to the propitious viewing geometry for sampling the tropopause fold.

Figure 6b documents the more complicated situation regarding the temperature perturbations due to the mountain waves along the GLORIA viewing direction. Here, HALO is located in a lobe of colder temperatures stretching from below the aircraft to above 100 hPa. Along the uppermost tangent points, the temperature perturbations vary remarkably in zonal direction within several tens of kilometres. The impact of the mountain waves can be also seen by undulations of the PFJ (Fig. 6c) as well as of the water vapour distribution above the Apennines (Fig. 6a), especially, at stratospheric levels higher than 200 hPa. Furthermore, the IFS vertical wind is shown in Figure 6d. The strongest updraft of about 0.8 m s$^{-1}$ is found around 14° E just below the aircraft (Fig. 6d). The vertical wind is in quadrature with the temperature anomalies: cold anomalies are associated with phase fronts of largest upward displacement and follow phase fronts of upward winds (adiabatic cooling) and warm anomalies are associated with the largest downward displacement after downward winds (adiabatic heating). The IFS data as presented in Figure 6 show that the temperature, wind and water vapour fields were influenced by vertically propagating mountain waves which interacted with the tropopause fold above the Apennines. According to the IFS data, the

GLORIA viewing direction was aligned across horizontal temperature contrasts approaching ±2 K within a few tens of kilometres in the upper part of the observations. Therefore, the comparison of the horizontally blurred GLORIA data (along viewing direction) with the model data sampled sharply at the tangent points can be strongly influenced by the observational filter. Further down at altitudes below the 220 hPa level, the IFS data show a more

homogenous temperature distribution along the GLORIA tangent points.

## 4 Comparison of GLORIA and in-situ data

Figure 7 displays both, a comparison of the GLORIA data at flight level with the BAHAMAS in-situ temperature data between way points A and D as well as the comparison of the retrieved vertical temperature and water vapour profiles with the LIRE 12Z (11 UTC) radiosonde sounding (for launch site see star marked in Figs. 4 and 5). Note,

no suitable flight level in-situ water vapour data is available for PGS06.

### 4.1 Flight level data

Figure 7a shows GLORIA water vapour (red) and the pressure at flight altitude (black). The grey lines mark the constant 10 ppmv values of the initial guess and a priori profile used for GLORIA's water vapour retrieval. Regions highlighted in yellow indicate where the GLORIA tangent points passed the radiosonde launch site ±50 km. Right

at way point A, GLORIA measured dry stratospheric air of about 3 to 5 ppmv at the cyclonic shear side of the PFJ. Further southward, water vapour mixing ratios increased gradually as HALO approached the tropopause layer. Water vapour varied at upper tropospheric mixing ratios between > 6 to 22 ppmv up to way point B. Thereby, the strongly oscillating signature of the observed water vapour might be related to the wave-induced up- and downdrafts along GLORIAs viewing paths (see Fig. 5a, b). At a distance of ~800 km, HALO had to lower the

flight level by ~10 hPa temporarily for safety reasons (see dip in pressure in Fig. 7a). Here, an unexpected feature is observed. While the water vapour mixing ratio of ~9 ppmv before the descent would be assumed to increase, the opposite is found: The water vapour mixing ratios drop back to stratospheric values of about 5 ppmv at minimum and rise again to ~9 ppmv as HALO climbed back to 180 hPa (FL410). Bramberger et al. (2018) explain this flight sequence in detail and GLORIA's water vapour observations can be interpreted by the abruptly changing

sign of the vertical wind induced by the mountain waves along HALOs flight track. Thus, GLORIA measured descending dry stratospheric air related to the mountain waves at this location. The subsequent ascent of HALO to FL430 (~160 hPa) after way point B brought the observed water vapour mixing ratios back to stratospheric levels. The amplitude of the water vapour oscillations along the northbound leg are much smaller since HALO was flying entirely inside stratospheric air masses.

Figure 7b shows the comparison of GLORIA temperature data (red) with the BAHAMAS temperature data (blue) along the flight track. The ECMWF HRES temperature data used as initial guess and a priori information for the GLORIA temperature retrieval is superimposed as a grey line. The smooth HRES profile follows the BAHAMAS data roughly indicating that the HRES data represent the large-scale meridional temperature variations very well.

Likewise, GLORIA's retrieval results follow the in-situ observations very closely. However, the GLORIA measurements reveal stronger temperature variations and more scattering than the HRES profile. In some flight sections (e.g., 900 km to 1400 km and 3300 km to 3700 km), the GLORIA results clearly resemble the BAHAMAS data more closely than the much smoother HRES data.

However, there are moderate deviations between the GLORIA and the BAHAMAS data which can be explained
by the different regions sampled, since the horizontal focus of the GLORIA observations at flight altitude is located about 50 km away from the carrier. For example, the negative offset of GLORIA temperature by 1-2 K compared to BAHAMAS directly south of way point A could be produced by zonal temperature gradients induced by the mountain waves whose phases are nearly aligned parallel to the flight track, cf. Fig. 5c and d. Moreover, a strong anti-correlation is seen between the GLORIA and BAHAMAS data during the first crossing of the radiosonde
launch site. There, the BAHAMAS temperature drops by more than 5 K within several tens of kilometres, while the GLORIA temperatures rise by 2-3 K at the same time. This anti-correlation can be explained by the mentioned phase orientation of the mountain-wave induced temperature perturbations. While HALO streaked along a strong local temperature minimum, the GLORIA temperatures observed remotely were dominated by a wave-induced, local temperature maximum located west of the flight track.

**4.2 Radiosonde sounding**

Figures 7c to 7f compare the GLORIA profiles with the LIRE radiosonde sounding at 11 UTC during the first and second HALO passage of the launch site. All GLORIA profiles from the two segments marked in yellow in Fig. 7a and b are considered for the comparison. In case of water vapour (Fig. 7c and 7d), the constant 10 ppmv initial guess and a priori profile is also shown. As background reference for temperature, again the initial guess and a
priori HRES profiles are added (Fig. 7e and 7f). While the geographical match of GLORIA tangent points covering the launch site was better during the first passage (Fig. 7c,e), the time period of the second passage suits better to the sounding (Fig. 7d,f; GLORIA data measured shortly after 11 UTC). However, during the second passage, GLORIA was pointing away from the launch site, and a v-shaped gap is found in the tangent point distribution due to a slight azimuth turn of HALO and a data gap. The vertical resolution of the GLORIA profiles as derived
from the traces of the usual 1D averaging kernels of the profiles are shown on the right hand side panels of Figures

7c to 7f. They amount to 300 m to 600 m below flight altitude. Further down, the vertical resolution deteriorates towards about 900 m. Above the flight level, the vertical resolution decreases rapidly due to the sampling characteristics of the airborne limb observations.

Below the flight level and above 11 km altitude, both the GLORIA and radiosonde data show typical dry
stratospheric water vapour mixing ratios which increased markedly to tropospheric values below 10 km (Figs. 7c, d) as HALO was located south of the tropopause break, see Fig. 3b. The sounding indicates $H_2O$ mixing ratios of around ~100 ppmv down to 7 km whereas the GLORIA profiles measure larger values up to 800 ppmv. During the first passage, all the GLORIA profiles show qualitatively the same altitude dependence (Fig. 7c). Both GLORIA and the sounding detected the strong decrease of water vapour associated with the stratospheric intrusion
at about 6 km altitude. During the second passage, however, two different GLORIA profiles are found (Fig. 7d). While one branch (red) follows the radiosonde data closely, the second branch (green) remains at very low stratospheric water vapour levels at all altitudes. The absence of a smooth transition between the two branches can be explained by a data gap and slight azimuth turn of HALO. As a consequence of both, GLORIA's viewing direction changed and horizontally separated air masses with different humidity were sampled. Unlike the
GLORIA measurements, the sounding data indicate another dry stratospheric intrusion as water vapour minimum at around 5 km altitude.

The comparison of the GLORIA temperature profiles with the radiosonde data are shown in Figs. 7e and 7f. The temperature sounding shows a vertical profile where the tropopause is characterized by a pronounced inversion at about 12 km, the tropopause inversion layer (TIL). GLORIA observations reproduce the TIL as well as the upper
tropospheric lapse rate very well during the first passage on the southbound leg. During the second passage, as for the water vapour profiles, two profile branches are visible in the GLORIA data: one set of profiles (red) follows closely the sounding below 12 km altitude and shows a sharp TIL. The other one (green) associated with the dry air masses exhibits a lower lapse rate and temperatures lower by 10 K at 8 km. This suggests that dry descending stratospheric air within the tropopause fold is responsible for this finding. Furthermore, at the height of the $H_2O$
minimum, the radiosonde sounding detected shallow layers of stably stratified air (Figs. 7e, f). The coincidence of local inversions with low water vapour values is in agreement with the conceptual picture of stratospheric intrusions into the troposphere as mentioned in the Introduction. Unfortunately, no GLORIA data exist at these lower levels.

## 5 Results

### 5. 1 Overview

Figures 8a, 8b, and 8e overview the water vapour, temperature, and ozone distributions observed by GLORIA between way points A and D. The tropopause fold can be identified by extruding dry and ozone rich air extending
down and southward from the stratosphere into the troposphere. The tongue-like extensions of stratospheric air are located between 300 km and 800 km during the southbound as well as between 2700 and 3300 km during the northbound leg (Fig. 8a, 8e). Within the tropopause fold, narrow bands of dry filaments with enhanced ozone values suggest advective processes as the main driver of their formation. Both at the upper north and lower south edges of the tropopause fold, gentle horizontal $H_2O$-gradients point to diffusive processes levelling the
troposphere-stratosphere contrast out. Temperature observations across the tropopause fold confirm the expected reversal of the meridional temperature gradient from the lower troposphere to the upper troposphere which is responsible for closing the jet stream in the lower stratosphere (Fig. 8b). South of the tropopause fold at the anticyclonic side of the PFJ, GLORIA observed a rather gradual, diffusive transition of high tropospheric to very low stratospheric $H_2O$-values between 300 hPa and 180 hPa. At certain places, high $H_2O$-values seem to be injected
into stratospheric altitudes. Interestingly, the low ozone values in this part of the flight suggest that the air is mainly transported upwards, there are only a few segments where ozone mixing ratios are enhanced indicating mixing processes between the stratosphere and troposphere. We refer to this layer as extratropical transition layer (ExTL), Gettelmann et al. (2011), as it marks the crossover from tropospheric to stratospheric air. Moreover, the presence of rather cold stratospheric spots with T < 215 K (maybe wave-induced) and clouds (blocked areas in Fig. 8a, b,
and e) point to non-adiabatic processes such as radiative cooling and cloud formation.

The corresponding IFS cross sections of water vapour and temperature taken at 10 UTC and interpolated to the GLORIA tangent points reveal already a remarkable agreement with the GLORIA observations (Fig. 8c, d). The tropopause fold is identified at the same position as in the GLORIA observations and its vertical and horizontal extents are similar to the observations. However, the $H_2O$-values inside the fold are higher and the observed
separated dry filaments are missing in the IFS data. Although the structure of the tropospheric water vapour distribution is well represented by the IFS, there are notable deviations to the GLORIA observations. Especially, the IFS simulates much smaller vertical $H_2O$-gradients in the transition zone from tropospheric to stratospheric air near the tropopause. Furthermore, higher water vapour mixing ratios reach up to higher altitudes and the very dry stratospheric air sensed by GLORIA north of PFJ is only partly reproduced. The comparison of GLORIA's
temperature with the IFS field shows a remarkable agreement (Fig. 8b, d). Again, local deviations occur. For

example, the observed temperature maximum around flight altitude at a distance of about 700 km and the cold spots south of the tropopause fold are reproduced moderately by the IFS.

## 5.2 Mesoscale fine structure

The IFS data used to compare with the GLORIA observations as shown in Figure 8 were interpolated directly to the tangent points. However, in the presence of horizontal $H_2O$ and temperature gradients and mesoscale fine structures along GLORIA's line-of-sight view, the horizontal smoothing characteristics of GLORIA may affect the comparison significantly. Therefore, we apply the observational filter in the following to sample the IFS data as outlined in Sec. 2.3. Figures 9a and 9b display zooms of the retrieved water vapour distributions along the tropopause fold during the south- and northbound legs, respectively. In the same way, Figs. 9e and 9f show the ozone distributions.

Essentially, Figure 9 juxtaposes snapshots of a tropopause fold at two different stages of its evolution which are separated by about 2.5 h and sampled in different viewing directions. GLORIA water vapour and ozone measurements show remarkably well-resolved signatures of stratosphere-troposphere exchange: Tongues of dry and ozone-rich stratospheric air intrude deeply into the troposphere (Figs. 9a, 9e). The observed distribution of the potential temperature reveals descending isentropes in agreement with the conceptual view of the processes inside the tropopause fold. Simultaneously, moist tropospheric air with low ozone values is entrained into the stratosphere. This happens in the vicinity of and above the core of the PFJ above about 280 hPa where the isentropic surfaces are folded and indicate mixing processes (Figs. 9a, e), see Sec. 5.4. In contrast, the sharper horizontal $H_2O$-gradients at the southern and lower side of the tropopause fold are indicative of advective processes extruding dry stratospheric air down. During the second passage, essentially the same overall structure of the tropopause fold was observed (Figs. 9b, 9f). However, the intruding filaments of low $H_2O$ and high ozone are much fainter inside the fold. Furthermore, the water vapour distribution in the jet stream region appears notably smoother. Again, a sharp contrast towards tropospheric water vapour is found in the lower compartment of the fold at the anticyclonic side. (Fig. 9b). The GLORIA observations as presented in Figs. 9a and 9b and Figs. 9e and 9f are the first combined temperature and trace gas observations of an horizontally and vertically extended active mixing region belonging to a tropopause fold.

Figures 9c and 9d juxtapose the IFS water vapour fields using the respective observational filters $\widetilde{A}$ at 09 UTC and 11 UTC, respectively. The application of the observational filters improves the agreement with GLORIA observations significantly (see Appendix A). Overall, the IFS water vapour distributions agree remarkably with

the GLORIA observations. The tropopause fold is wrapped around the jet stream and is found at the same position as in the observation and its overall shape compares well to the GLORIA measurements. However, the IFS fields appear being too moist in the stratosphere and the very low $H_2O$-values as observed by GLORIA were not reproduced. Furthermore, a smoother transition from tropospheric towards stratospheric mixing ratios is found in the IFS data. In spite of the rather high spatial resolution of the global IFS data, the filamentary structure inside the tropopause fold is not resolved.

### 5.3 Mountain-wave induced temperature perturbations

Figures 10a and 10b display the observed temperature perturbations (versus IFS background temperatures, see Sect. 2.2) and the potential temperature along the south- and northbound legs, respectively. During the first passage of the tropopause fold, the GLORIA observations show large temperature perturbations exceeding ±3 K, whereas their values are slightly lower and mostly do not exceed ±2 K during the second passage on the northbound leg. Below about 300 hPa, the temperature perturbations reflect the large-scale meridional temperature gradient discussed above in relation to Fig. 8b: warm anomalies are located to the south, cold anomalies to the north. Above that level, the meridional gradient of absolute temperature reverses sign (Fig. 8b) and this is generally reflected here by warm temperature anomalies in the north and cold ones to the south (Fig. 10a and b). Moreover, there are alternating patterns of positive and negative temperature perturbations above about 300 hPa which indicate the presence of mountain waves. On the southbound leg, their phase lines extend from 45°N to about 41°N whereas on the northbound leg they are restricted to the segment from 40°N to 42°N when HALO approached the Italian peninsula from the south again (Fig. 5).

As mentioned above, HALO flew nearly parallel to the phase lines of the vertically propagating mountain waves. This was confirmed by the IFS phase line orientations as documented in Fig. 5 and also by recent high-resolution numerical simulations presented in Bramberger et al. (2018), their Fig. 11. Therefore, the observed tilt of the phase lines and their changing vertical wavelength are related to the different propagation conditions in terms of wind and stability along the flight track. Especially, the temperature perturbations along the southbound leg show nearly horizontal orientation inside the core of the PFJ at around 42°N. In this segment the largest temperature fluctuations were observed and they are related to the sudden and dramatic change of temperature and meridional wind at flight level which resulted in the stall event and the subsequent forced descent of HALO (Bramberger et al., 2018).

The vertical wind of the IFS short-term forecasts provides indication of the secondary circulation around the core of the PFJ. Whereas the vertical velocity along the northbound leg (Fig. 10d) reveals nearly undisturbed upwelling

south and downwelling north of the PFJ as expected, the vertical flow field is heavily disturbed along the southbound leg (Fig. 10c). There again, stacked alternating positive and negative values indicate the presence of mountain waves.

In the IFS data, the temperature perturbations mainly reflect the large-scale meridional temperature gradient (Fig. 10e and f). As shown in Appendix A, particularly for temperature the application of the observational filter strongly improves the agreement with the GLORIA observations. Their amplitudes are, however, generally weaker than in the GLORIA observations. Exceptionally, the vertical structure of the temperature fluctuations north of the stall event around 500 km is well represented, indicating that the IFS captured the vertically propagating mountain waves at least partially. Overall, the IFS temperature perturbations reproduce the major structures found in the GLORIA data well. Temperature amplitudes are weaker by typically ~1 K, and less fine structures are identified.

**5.4 Mixing in the vicinity of the tropopause fold**

The panels (a) – (d) of Figure 11 show the tracer-tracer correlations of GLORIA observations between 40°N and 45°N as function of potential temperature in different views. These plots include simultaneous GLORIA observations of water vapour, ozone, and potential temperature covering an altitude range of about 6 km below HALO's flight level. As expected, higher ozone values are correlated with low $H_2O$-values and characterize the stratospheric tracer distribution along the vertical axis. Conversely, low ozone and high $H_2O$-values represent the tropospheric tracer distribution along the horizontal axis, Figs. 11a and b. In general, the potential temperature follows the altitude dependence, i.e. lower $\Theta$-values are associated with high/low $H_2O/O_3$-values and vice versa. Maximum water vapour values appear at the intermediate range of $310\,K < \Theta < 330\,K$, marking high tropospheric mixing ratios sampled south of the tropopause fold (compare Fig. 11b with high mixing ratios and isentropes south of the tropopause fold in Figs. 11e and 11f). Low $H_2O$-values in Figures 11a to 11d around $\Theta = 310\,K$ and $320\,K$ correspond to the stratospheric water vapour mixing ratios within the lower compartment of the tropopause fold, while the slightly enhanced $H_2O$-values (up to ~50 ppmv) below $\Theta = 310\,K$ mark the upper troposphere at the north of the tropopause fold (cf Fig. 11b, 11e and 11f).

Classically, the tracer relationship is used to define the provenance of the sampled air masses. Gettelman et al. (2011) used $H_2O$ mixing ratios of less than about 12 ppmv as threshold for stratospheric air and ozone values of less than about 65 ppbv as threshold for tropospheric air. The respective thresholds are added in Figs. 11a, c and d. This means, data points which are close to both tracer axes of the diagrams in Figs. 11a to 11d indicate an atmospheric state with no mixing and belonging either to the stratosphere or the troposphere, e.g. Plumb (2007)

and Gettelman et al. (2011). Data outside these regions are called mixed regions of the ExTL. Typical aircraft observations of stratosphere-troposphere mixing along vertically stacked flight legs appear in such diagrams as so-called mixing lines connecting both reservoirs; see Fig. 11 in Gettelman et al. (2011). Here, the zoom into the GLORIA observations along the individual legs reveals a wide-spread mixing area without individual mixing lines.

These observations indicate active stratosphere-troposphere exchange in the vicinity of the tropopause fold. Especially between 330 K and 350 K, we find enhanced water vapour values, which are accompanied by notably enhanced ozone volume mixing ratios above 200 ppbv (Fig. 11b, c and d). On the other hand, slightly enhanced ozone values up to 200 ppbv, which are accompanied by enhanced $H_2O$ well above 20 ppmv are found particularly below 330 K. The change of the shape of mixing region with potential temperature is visualised in Figure 11b by

colour-coding only data points falling into the 2D mixing area.

To illustrate the locations of the mixing regions, data points indicative for the ExTL mixing region (water vapour > 12 ppmv and ozone > 65 ppmv) are flagged in the vertical cross section displayed in Figures 11e and 11f. Particularly, the data points with Θ-values between 330 K and 350 K, sticking out most of all in the correlations (data points with ozone up to 400 ppbv and more at water vapour well above 12 ppmv), show compact distributions

above the PFJ core which are tilted towards the fold. This suggests that the secondary circulation around the core of the PFJ (cf. Fig. 10c and 10d) entrains moist air from the troposphere at the anticyclonic side and mixes it into the stratosphere. Vice versa, the flagged data points characterized by lower Θ values below the PFJ core in Fig. 11e indicate mixing of stratospheric air masses into the troposphere due to advection and filamentation, probably intensified by the PFJ secondary circulation. Here, and also in the ExTL mixing region south of the tropopause

fold, only weaker enhancements of ozone up to ~200 ppbv are found for the data points attributed to the mixing zone. In Fig. 11f, the dense mixing region in the lower compartment of the fold is missing probably due to the data gap between 42°N and 43°N. Figure 11b reveals how the mixing region changes, like on a spiral stair, from a predominantly tropospheric correlation (low ozone, strongly variable water vapour) below 330 K to a predominantly stratospheric correlation (low water vapour, more variable ozone). Thereby, the 330 K isentrope

roughly separates the regions of stratosphere-to-troposphere and troposphere-to-stratosphere exchange.

The ExTL mixing zone during the first passage of the tropopause fold is notably disturbed by the mountain wave (cf. Fig. 11e). The wave-induced temperature anomaly (Fig. 10a) results in a distortion and spreading of the isentropic bands in the mixing zone between 41.5°N and 43°N above 270 hPa. This coincides with the clear air turbulence mixing region in the vicinity of the jet stream core (Shapiro, 1980) and suggests that ongoing mixing

processes are altered. Moreover, the temperature anomaly results in a steepening and overturning of isentropes.

The consequence is a region of convective instability, which further supports turbulence and mixing. When compared to the mixing zone associated with a tropopause fold near the subtropical jet stream in July 2006 analysed by Ungermann et al. (2013), a larger mixing region stretching deeper into the jet stream core is observed here. The large extent of the observed mixing zone thus might be a consequence of the interaction with the mountain wave.

Another interesting detail is found when Figures 10c and 11e are compared: the separated narrow moist filament attributed to mixing zone between ~43.5°N/250 hPa and ~42.5°N/320 hPa (Fig. 11e) coincides with the position and alignment of a region of wave-induced wind shear (Fig. 10c). While advection within the tropopause fold probably is the main driver for the formation of the filaments, the wave-induced wind shear is likely to enhance mixing processes here. Further complex patterns of the data points at the lower southern side of the PFJ attributed

to the mixing zone are interpreted as being produced by the complex dynamics and mixing processes around the PFJ.

**6 Conclusions**

This paper reports findings of remote-sensing observations of a tropopause fold by means of the GLORIA instrument aboard the German research aircraft HALO over northern Italy on 12 January 2016. The design of the

chosen flight tracks crossing the PFJ meridionally provided the opportunity to observe the deep intrusion of stratospheric air in a tropopause fold as well as mountain waves in great detail. In this way, the particular characteristics of the linear GLORIA observations were used to their best advantages: the high-spectral resolution along the flight track enabled accurate observations of the predominant large meridional gradients of temperature and trace gases whereas the lower resolution normal to the flight track sampled air with smaller spatial variability.

In this particular configuration, the GLORIA high-spectral resolution measurements are advantageous and extremely valuable as they provide simultaneous vertical profiles of temperature, water vapour and ozone, and potential temperature with high definition.

Our findings comprise the detection of a broad intrusion of dry and ozone-rich stratospheric air. The combination with the IFS wind confirms the accepted conceptual model of a tropopause fold associated with the PFJ.

Particularly, we detected narrow filaments of water and ozone inside the tropopause fold at the cyclonic shear side of the PFJ that indicate advective processes predominantly forming the stratospheric intrusion. The astonishing detection of such fine filaments by GLORIA was only possible due to the high spatial resolution along the flight track. These observations confirm earlier findings of the stably stratified laminae extruding stratospheric air down to the troposphere (Shapiro et al. 1987). Furthermore, broader tongues of moist air entraining tropospheric air were

observed at the southern, anti-cyclonic shear side of the PFJ. These GLORIA observations in combination with the IFS wind suggest entrainment of moist tropospheric air into the stratosphere by the secondary circulation around the PFJ core. Due to the flow across the Apennines and the nearly parallel alignment of lower and upper tropospheric winds, the tropopause fold was perturbed by vertically propagating mountain waves. The wave

signatures are clearly evident in the GLORIA temperature field, and the tracer field shows a displacement of moist tropospheric air deep into the ExTL mixing region above the PFJ. Moreover, the wave-induced temperature modulations found in the GLORIA data approach amplitudes of $\pm 3$ K and result in spreading, steepening and overturning of isentropes in the PFJ core region, fostering vertical isentropic transport, mixing processes and possibly partial wave breaking.

In contrast to one-dimensional airborne in-situ observations, GLORIA's high-vertical resolution vertical profiles provide an almost complete 2-dimensional sampling of mixing in the vicinity of the PFJ. For this purpose, tracer-tracer correlations of water vapour and ozone were constructed. Here, the common mixing lines are replaced by mixing areas due to the dense vertical coverage of the GLORIA measurements. Specifically, the combination with the measured potential temperature presents a detailed 2D view of active mixing in the extra-tropical transition

region. Mixing takes place in the interval 300 K $< \Theta <$ 350 K, with stratosphere-to-troposphere exchange taking place predominantly below 330 K and troposphere-to-stratosphere exchange above. Two major locations of mixing were identified: a broad region on the anti-cyclonic shear side of the PFJ where warm and moist air ascends and entrains air into the stratosphere. The other active mixing region is located at the lower edge of the tropopause fold where vertical shear and differential advection leads to mechanical turbulence production.

Finally, the GLORIA observations validated the short-term forecasts of the high-resolution IFS. The agreement of the overall features of the tropopause fold as location, shape as well as vertical and horizontal extents is astonishing. Naturally, the IFS cannot reproduce the sharp gradients at the edges and the filaments inside the tropopause fold. Furthermore, the fine structure of the mixing region was not reproduced even with the used 9 km horizontal resolution of the IFS, and a smoother transition of high tropospheric water vapour mixing ratios into the

stratosphere was found at the anti-cyclonic shear side of the PFJ. The application of GLORIA's observational filters instead of interpolation of the model data directly at the tangent points clearly improves the agreement with the remote-sensing data. Overall, the combination of the GLORIA and IFS data provides a detailed view of a tropopause fold, resolves an active mixing region, and suggests that mountain wave perturbations have the potential to modulate exchange processes in the vicinity of tropopause folds.

**Data Availability**

The GLORIA remote sensing and BAHAMAS in situ data are available at the HALO database via https://halo-db.pa.op.dlr.de. The ECMWF IFS and HRES data are freely accessible at https://www.ecmwf.int/. The radiosonde data was accessed at the Wyoming Atmospheric Soundings website (Department of Atmospheric Science, University of Wyoming, USA, http://weather.uwyo.edu/upperair/sounding.html).

**Appendix A**

Figure A1 shows the comparison of the GLORIA water vapour observations during both tropopause fold passages (a,b) together with the IFS data sampled at the GLORIA tangent points (c,d) and using the GLORIA observational filters $\tilde{A}$ (e,f). The overall pattern and the absolute water vapour mixing ratios are very similar due to the propitious viewing geometry along the fold. However, several details show a better agreement with the GLORIA observations after application of the observational filter: higher mixing ratios are found in Fig. A1e between 450 and 550 km around 400 hPa, and the extent of the lower compartment of the tropopause fold to the south compares better with the GLORIA observations. Also during the second passage, the shape and extent of the lower compartment to the south agree better with GLORIA.

Figure A2 shows the corresponding temperature residuals derived from the GLORIA observations during both tropopause fold passages (a,b) together with the corresponding temperature residuals retrieved from the IFS data. Again, the IFS data is sampled at the GLORIA tangent points (c,d) and using the GLORIA observational filters $\tilde{A}$ (e,f). Here, the application of the observational filters strongly improves the agreement with the GLORIA observations. The IFS data sampled at the GLORIA tangent points shows hardly any agreement with the patterns found in the GLORIA data. In particular, during the change of the flight level (see Bramberger et al., 2018), the IFS data at the GLORIA tangent points shows negative temperature residuals in accordance with the in situ observations (cf Fig. 7b), while the GLORIA observations show a temperature maximum (see Sect. 4.1). The application of the observational filter resolves this apparent discrepancy (Fig. A2e), and a local temperature maximum as in the GLORIA data is found (Fig. A2a). Also the developed maximum-minimum structure in the GLORIA data between 450 and 600 km at flight altitude is reproduced remarkably well after application of the observational filter, while this structure is not found in the IFS data sampled at the GLORIA tangent points. Also during the second passage, improved agreement between the GLORIA and IFS data is found when the IFS data is sampled using the observational filter. While the IFS data at the GLORIA tangent points (Fig. A2d) show a notable

local temperature minimum at a distance of 2800 km not seen in the GLORIA data (Fig. A2b), this structure disappears after application of the observational filter (Fig. A2f).

The fact that only a moderate improvement of the agreement is found in the case of water vapour and a strong improvement in case of the temperature residuals can be understood by considering the zonal vertical cross-

sections of the IFS data shown in Figure 6. While in the case of water vapour (Fig. 6a) comparably homogeneous conditions were present along the GLORIA viewing direction, sequences of strong temperature modulations are found in the IFS data within a few tens of kilometres (Fig. 6b). Thus, sampling of the IFS data at single points may capture local extremal values, while the application of the observational filters (cf. Fig. 2) takes into account the horizontal smoothing intrinsic to the GLORIA limb observations and improves the quality of the comparison

considerably.

**Acknowledgements**

This work was supported by the German research initiative ROMIC (Role of the Middle Atmosphere in Climate) funded by the German Ministry of Research and Education (BMBF) project "Investigation of the life cycle of gravity waves" (GW-LCYCLE, 01LG1206A). This work was furthermore supported by the German Research

Foundation (Deutsche Forschungsgemeinschaft, DFG Priority Program SPP 1294). F. Haenel has received funding from the DFG project no WO 2160/1-1. S. Johansson has received funding from the European Community's Seventh Framework Programme (FP7/2007-2013) under grant agreement 603557. We are grateful to the GLORIA team and DLR-FX for performing the measurements and HALO flights during PGS. We thank Andreas Giez and the BAHAMAS-Team of DLR-FX for providing the BAHAMAS data. The campaign was

furthermore supported by the GWEX project (Technical Assistance for the Deployment of the GLORIA instrument during the Gravity Wave Experiment, ESA Contract No: 4000115111/15/NL/FF/ah). We thank the European Centre for Medium-Range Weather Forecasts (ECMWF) for providing the meteorological analyses. We acknowledge Wyoming Atmospheric Soundings (Department of Atmospheric Science, University of Wyoming, USA) for providing the radiosonde data (see http://weather.uwyo.edu/upperair/sounding.html). We acknowledge

support by Deutsche Forschungsgemeinschaft and Open Access Publishing Fund of Karlsruhe Institute of Technology.

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

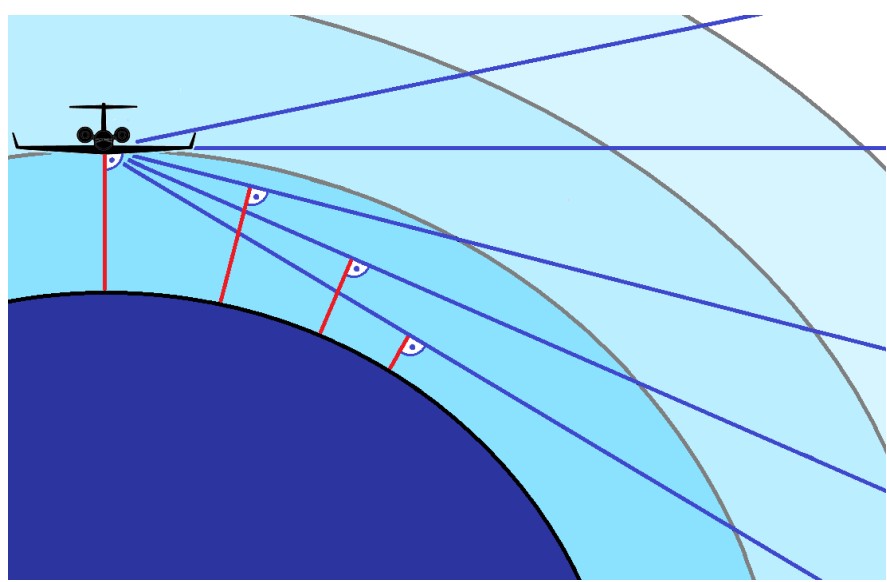

**Figure 1:** Illustration of GLORIA observation geometries (blue lines) with the carrier HALO moving away from the reader. Limb observations are characterized by their tangent points, where the line of sight is closest to the earth surface (red lines). Low tangent points are situated further away from the observer than high tangent points. The regions around the tangent points contribute the major part of the information derived in atmospheric parameter retrievals. Upward viewing observations have no tangent points along the line of sight and contribute limited information on the scenario above the flight track.

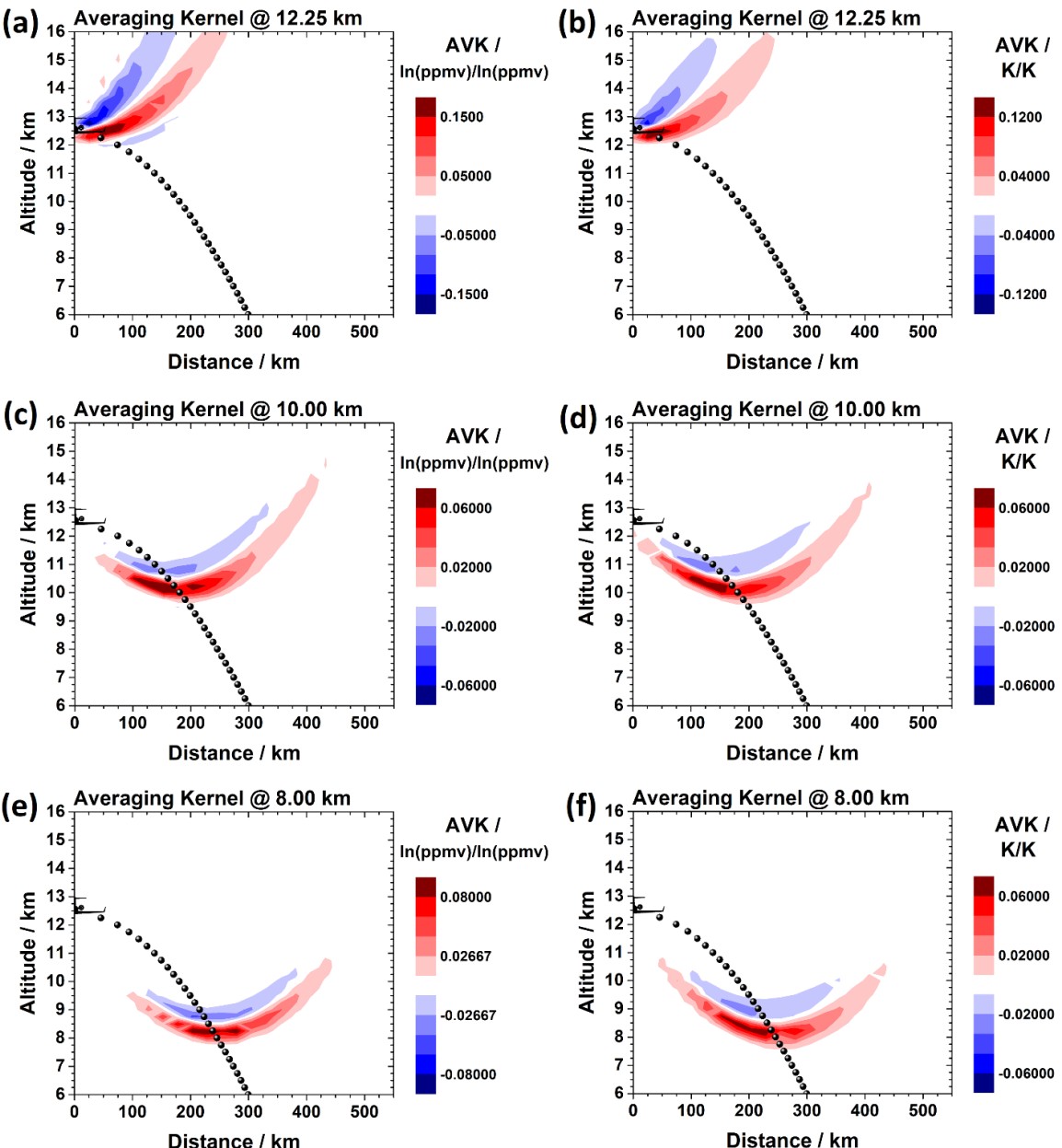

**Figure 2:** GLORIA observational filters including the horizontal domain along the line of sight. The plots show rows of the 2-dimensional averaging kernels $\widetilde{A}$ for the logarithm of water vapour (a,c,e) and temperature (b,d,f) corresponding with the retrieval grid levels indicated at the top of the panels. Black dots in all panels: retrieval grid tangent points of the GLORIA retrieval.

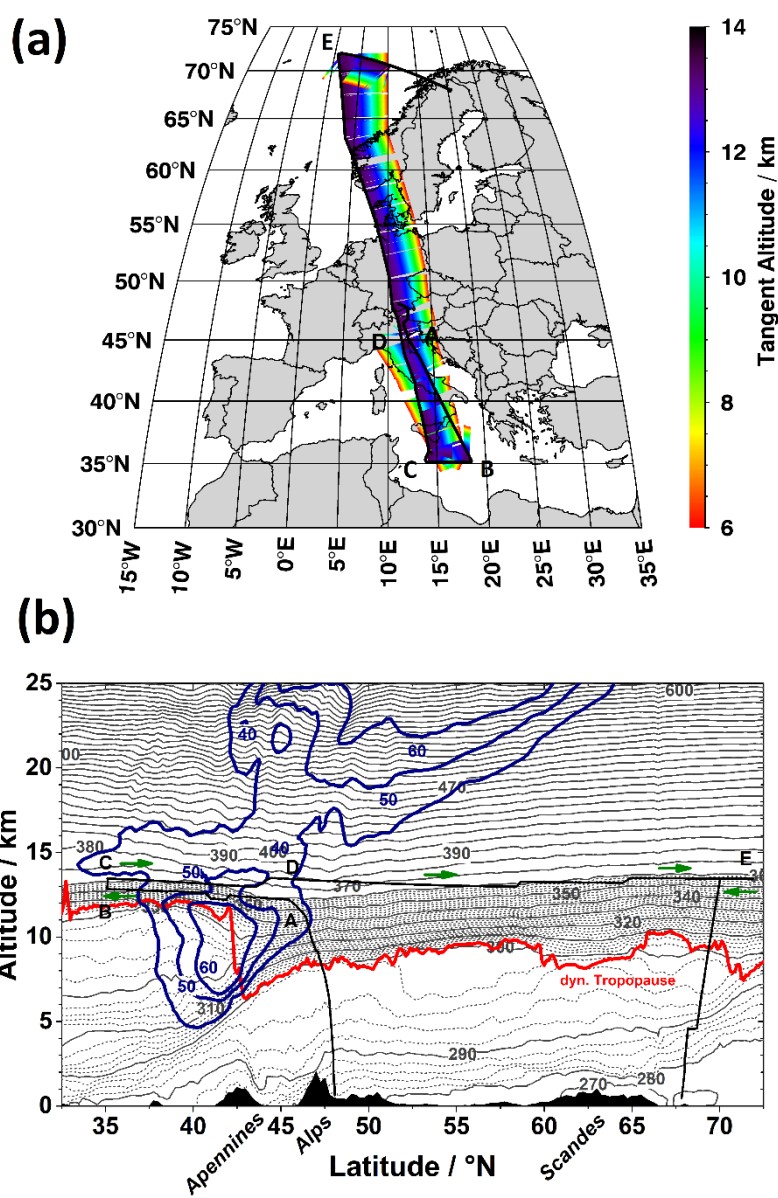

**Figure 3:** Horizontal and vertical profiles of the HALO flight PGS06 on 12 January 2016: (a) Flight legs (black
line) and tangent points of GLORIA observations, color-coded with tangent altitude in km. (b) Flight track (black
line) and topography (black). Green arrows indicate flight direction. Contours: Horizontal wind (m s$^{-1}$, blue lines)
and potential temperature Θ (K, ΔΘ = 10 K, solid lines and ΔΘ = 2 K, dashed lines up to Θ = 360 K) at 13°E
from ECMWF IFS valid at 10 UTC. Way points A, B, C, D, and E are mentioned in the text.

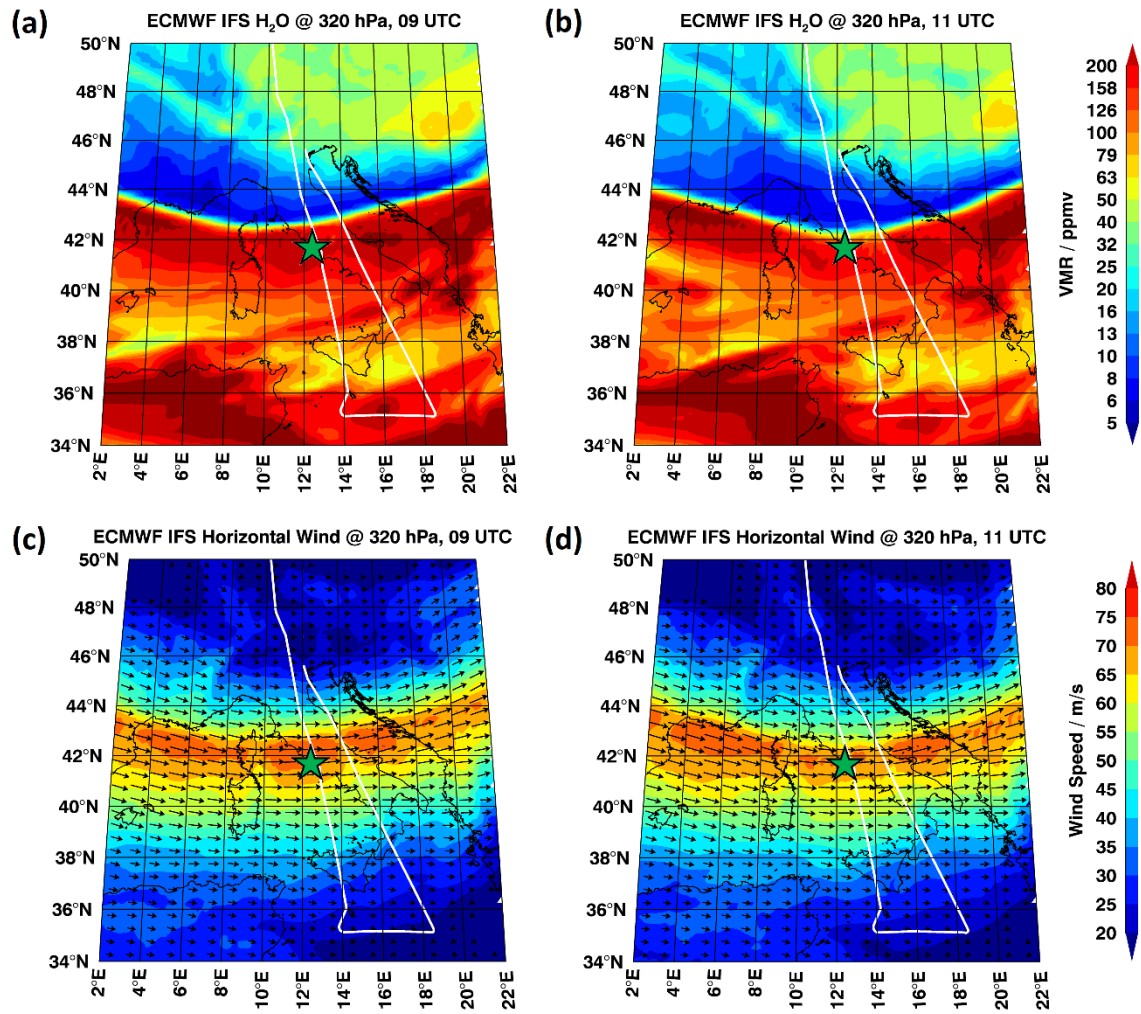

**Figure 4:** Horizontal section of water vapour mixing ratio (a, b) and horizontal wind (c, d) at the 320 hPa pressure surface at 09 UTC (a, c) and 11 UTC (b, d) on 12 January 2016. White lines: HALO flight track between ~08:25 UTC to 12:30 UTC. Green star in all panels: LIRE Pratica De Mare radiosonde launch site.

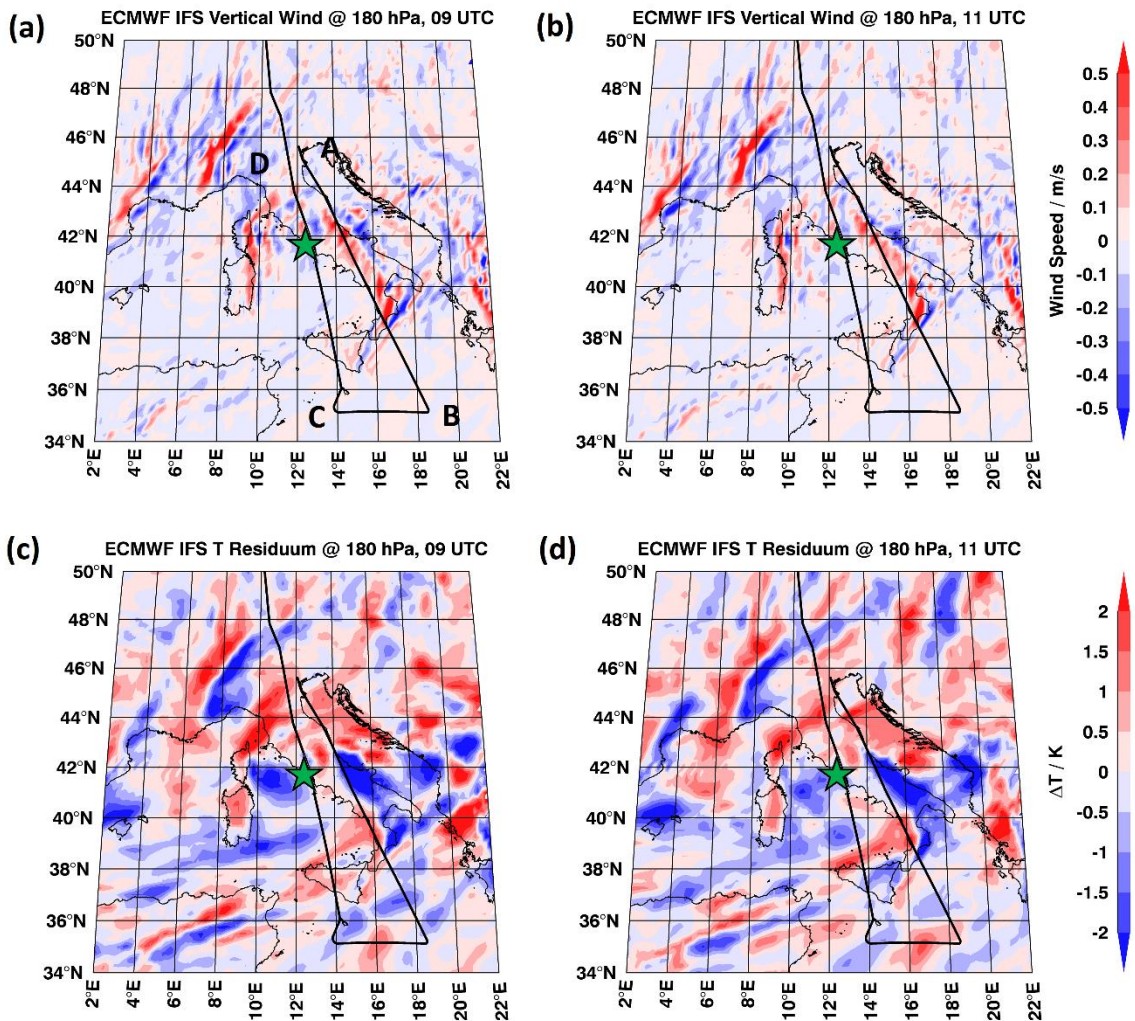

**Figure 5:** Mountain wave activity on 12 January 2016. Vertical wind (m s⁻¹, colour shading) at 180 hPa forecasted by the IFS a 09 UTC and 11 UTC (a,b). Temperature perturbations (K, colour shading) at 180 hPa forecasted by the IFS at 09 UTC and 11 UTC (c,d). Black lines: HALO flight track between ~08:25 UTC to 12:30 UTC. Green star in all panels: LIRE Pratica De Mare radiosonde launch site.

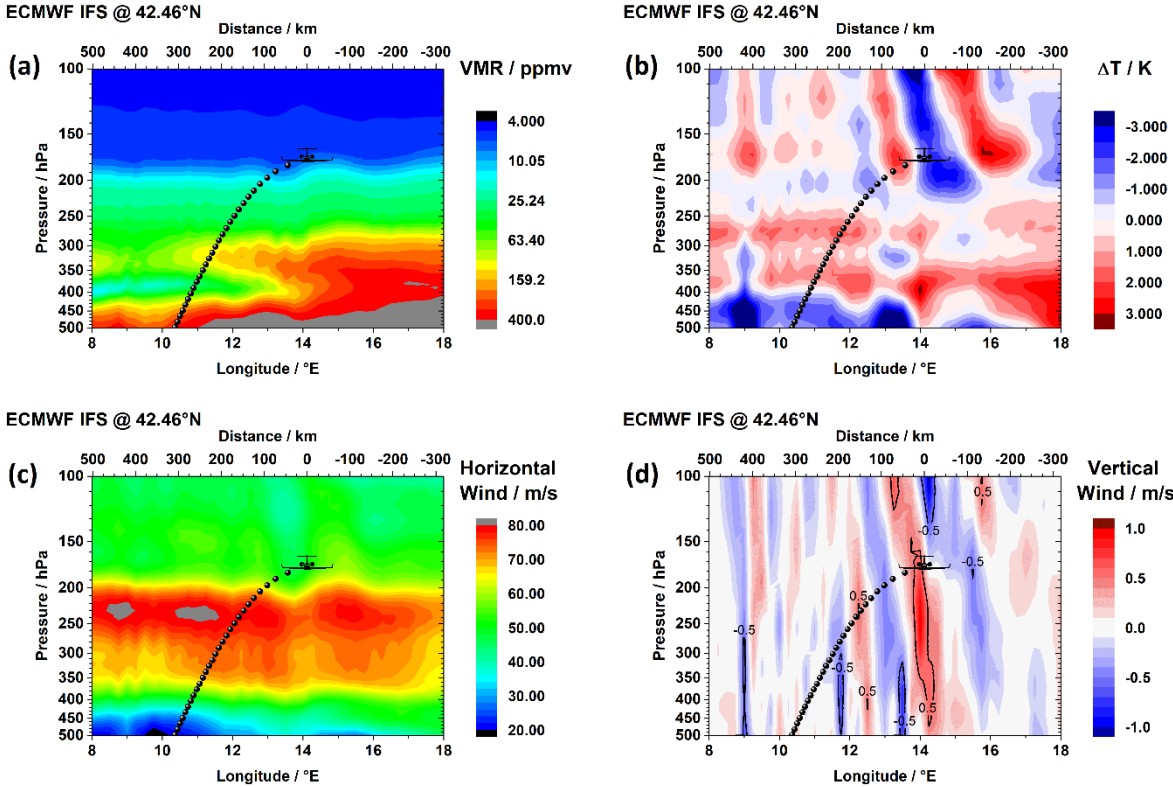

**Figure 6:** Vertical cross section of IFS water vapour (a), temperature perturbations (b), horizontal wind (c) and vertical wind (d). Black dots in both panels: projection of GLORIA tangent points (of data cube with uppermost tangent points coinciding with indicated latitude). Note that the GLORIA viewing direction was aligned not exactly in zonal direction, but slightly tilted to the south-west (see Fig. 3a).

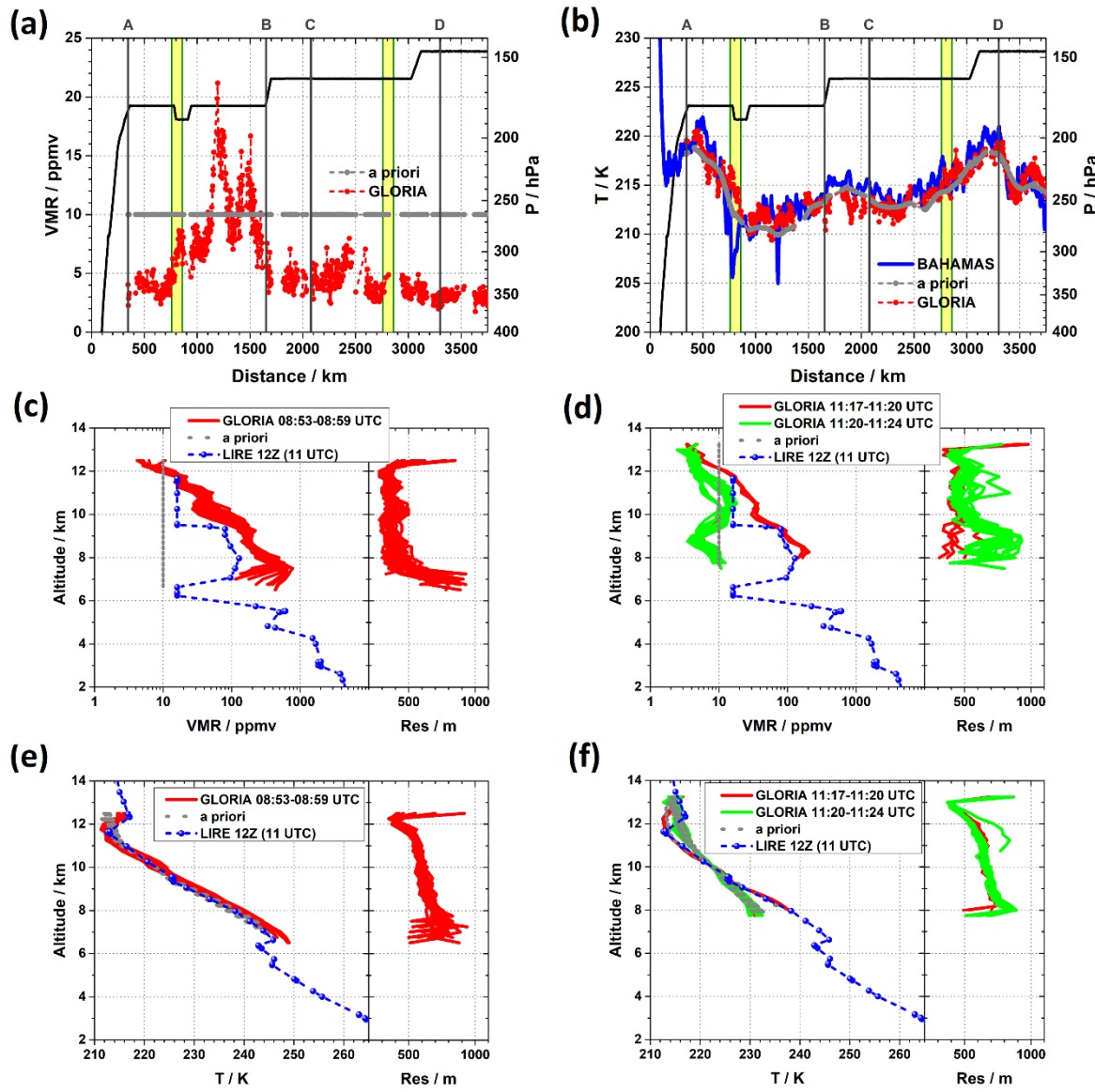

**Figure 7:** Comparison of GLORIA data with in situ data along flight track and radiosonde data. (a) GLORIA water vapour and initial guess/a priori data along flight track. (b) GLORIA temperature data, HRES initial guess/a priori data and BAHAMAS in situ data along flight track. Flight altitude in (a) and (b) is shown in black and refers to the right y-axis. Windows highlighted in yellow correspond with the passages of the radiosonde launch site. (c,d) GLORIA water vapour profiles and constant 10 ppmv initial guess/a priori profiles during first and second passage of the radiosonde launch site together with the radiosonde profile. (e,f) GLORIA temperature profiles and HRES initial guess/a priori profiles during first and second passage of the radiosonde launch site together with the radiosonde profile.

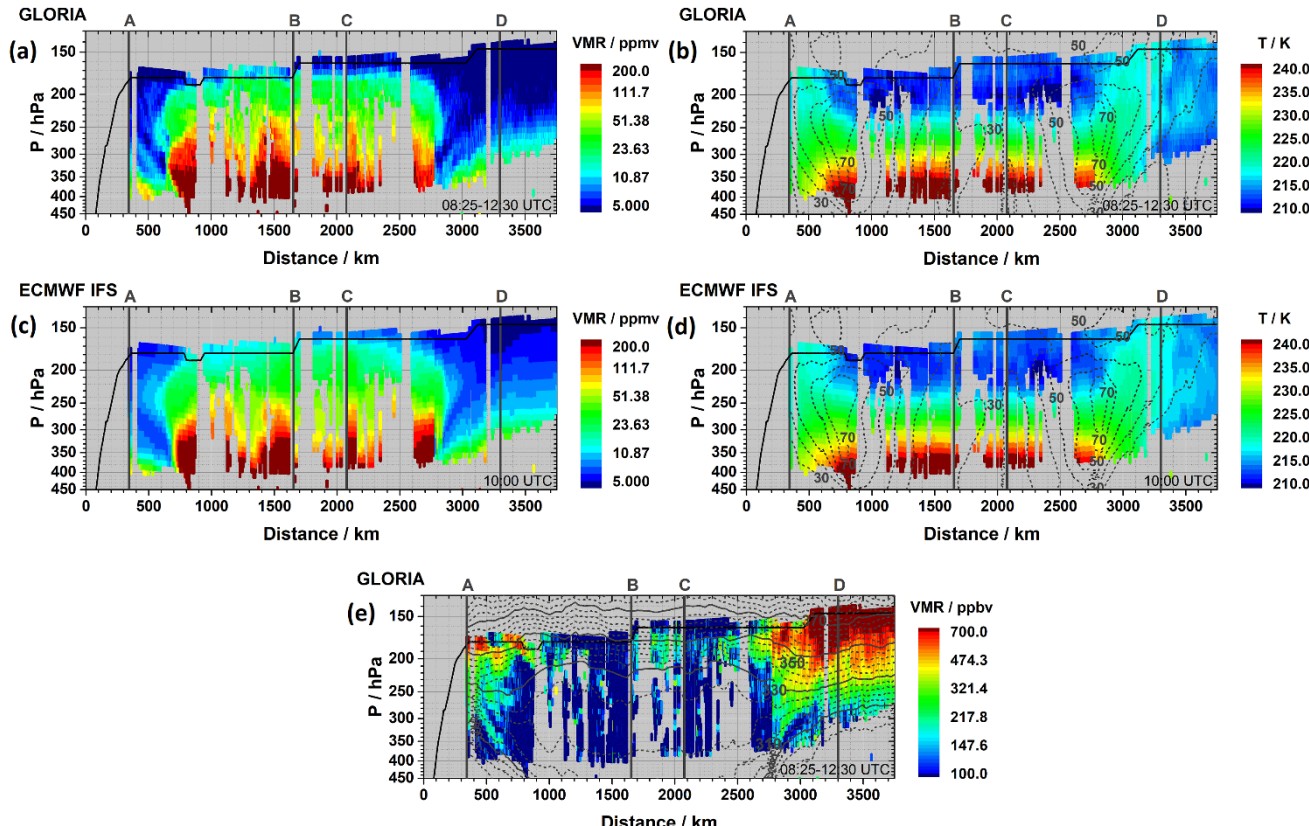

**Figure 8:** GLORIA observations of water vapour mixing ratio (a), absolute temperature (b), and ozone mixing ratio (e) along two passages of the tropopause fold. IFS water vapour mixing ratio (c) and temperature (d) interpolated to the GLORIA tangent points. IFS horizontal wind $V_H$ (m s$^{-1}$, dashed lines, $\Delta V_H = 10$ m s$^{-1}$) is superimposed on (b) and (d), potential temperature $\Theta$ (K, solid and dashed lines, $\Delta\Theta = 4$ K) on (e). Bold solid lines in all panels mark HALO's flight levels. Way points mentioned in the text are named by A, B, C, and D.

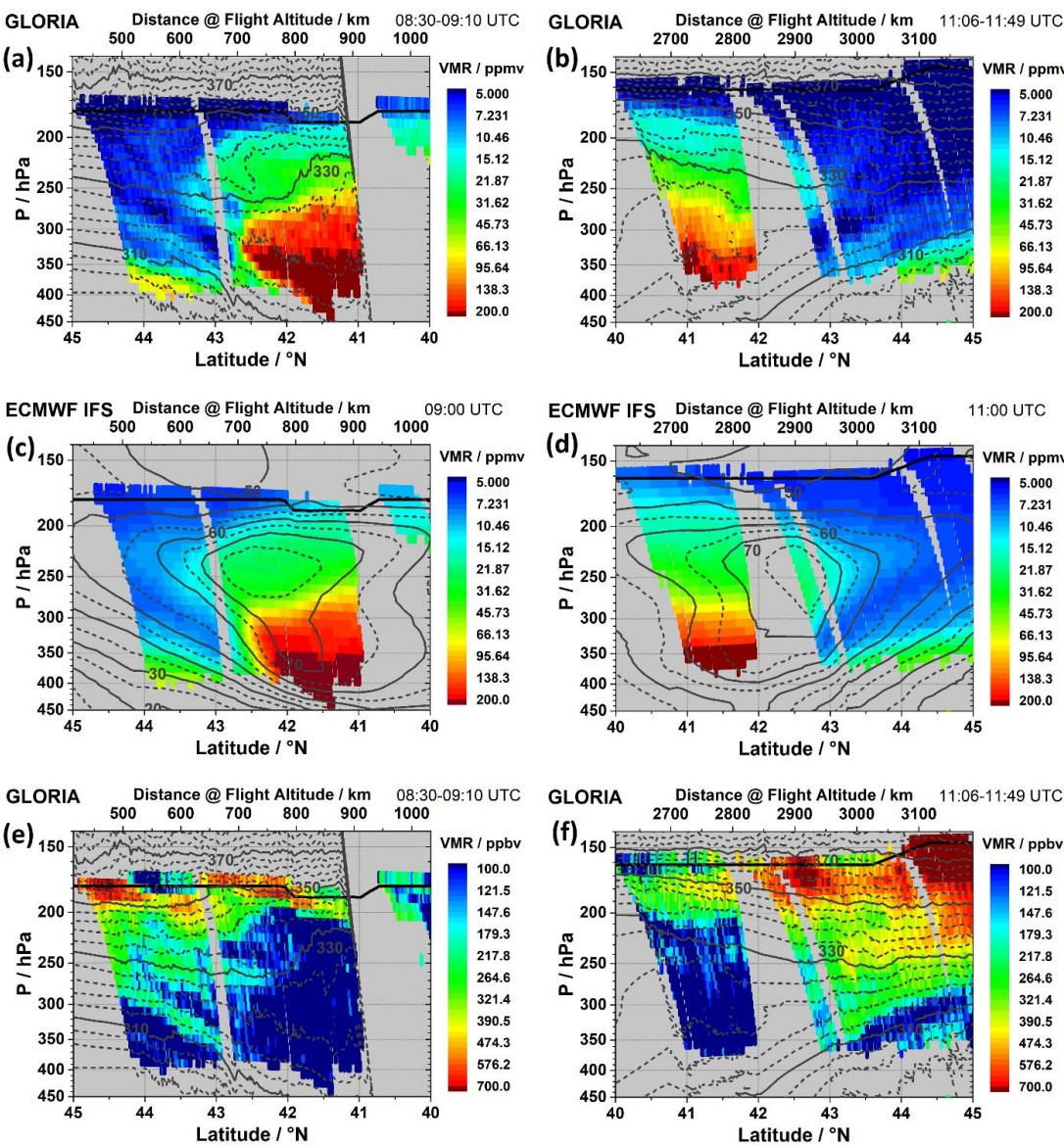

**Figure 9:** GLORIA observations of water vapour mixing ratio (a) and (b) and of ozone mixing ratio (e) and (f) for the southbound leg (left column) and the northbound leg (right column), respectively. Superimposed on panels (a), (b), (e), and (f) is potential temperature Θ (K, solid and dashed grey lines, ΔΘ = 4 K) derived from GLORIA temperature observations and HRES background pressure. Panels (c) and (d): IFS water vapour mixing ratio sampled using GLORIA observational filters. Horizontal wind $V_H$ (m s$^{-1}$, solid and dashed grey lines, $\Delta V_H$ = 5 m s$^{-1}$) is superimposed on (c) and (d). Bold solid lines in all panels mark HALO's flight levels.

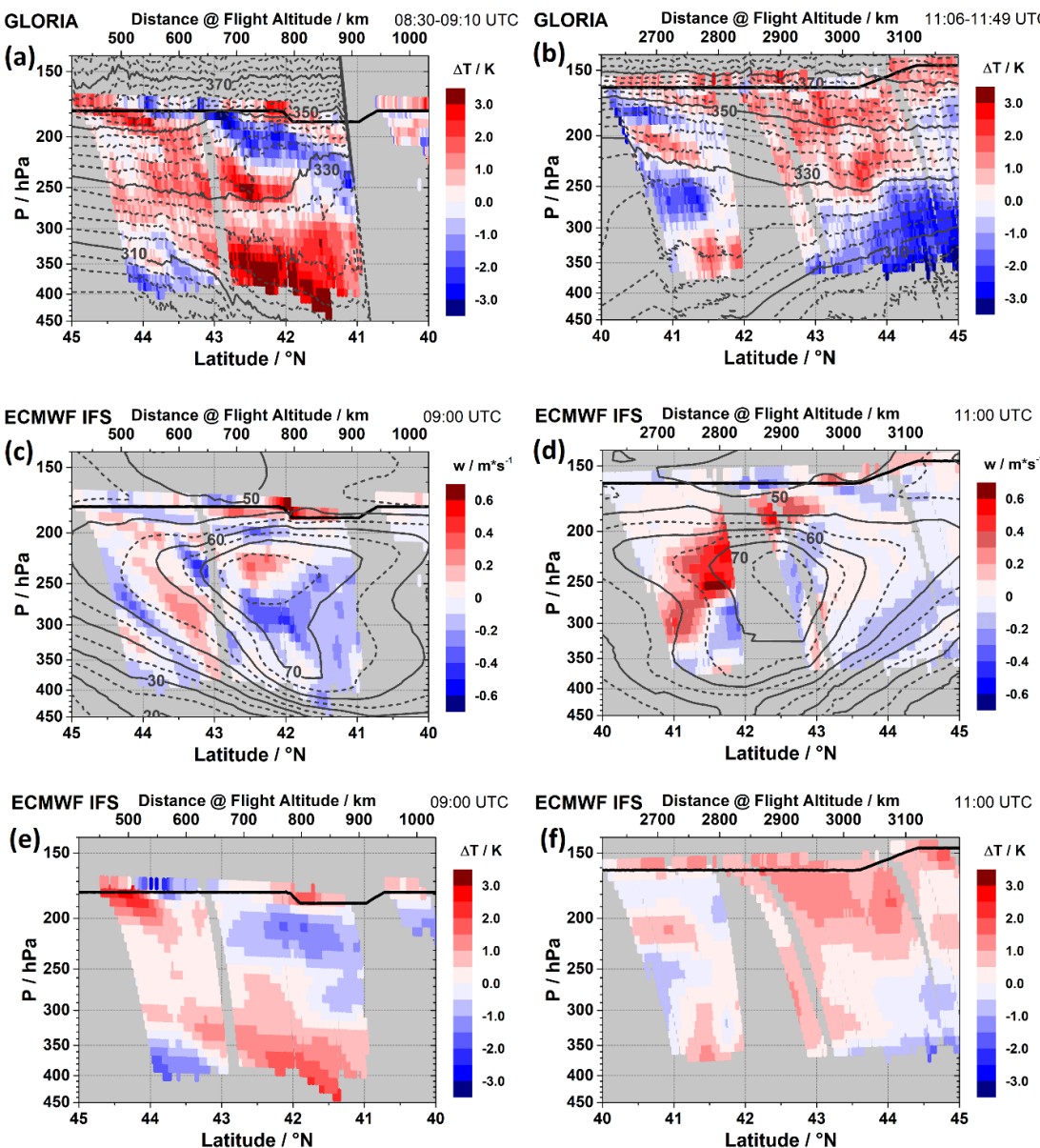

**Figure 10:** Temperature perturbations ΔT (K, colour coded) and Θ (K, solid and dashed grey lines, ΔΘ = 4 K) as derived from GLORIA's temperature measurements (a, b), IFS vertical wind (m s$^{-1}$, colour-coded) and $V_H$ (m s$^{-1}$, solid and dashed grey lines, $\Delta V_H$ = 5 m s$^{-1}$) at GLORIA tangent points (c, d), and IFS temperature perturbations ΔT (K, colour-coded) sampled using GLORIA observational filters (e, f) for the southbound leg (left column) and the northbound leg (right column), respectively. Bold solid lines in all panels mark HALO's flight levels.

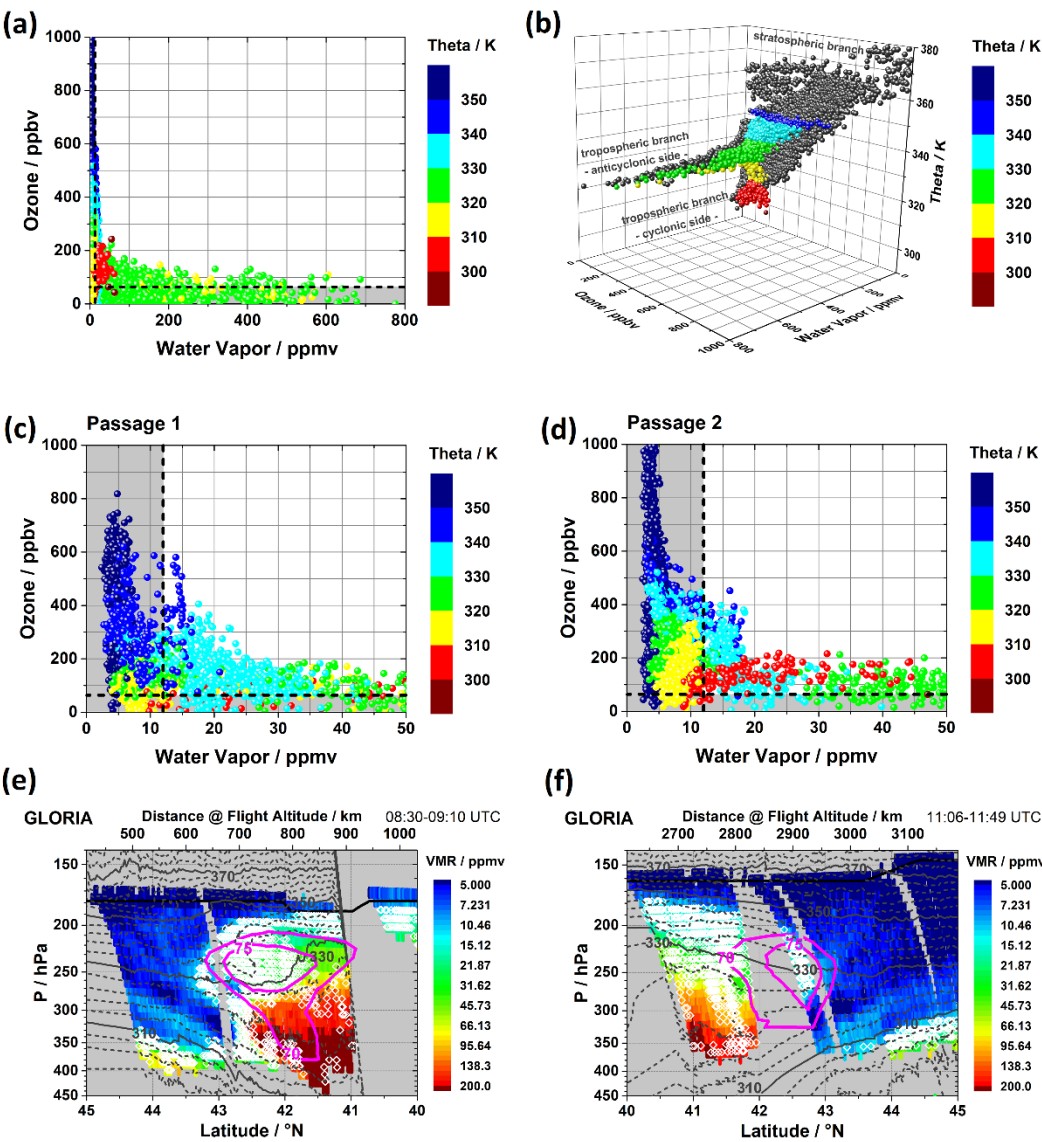

**Figure 11:** Tracer-tracer correlations of GLORIA's water vapour and ozone profiles during both tropopause fold passages as function of the observed potential temperatures. All observational data along south- and northbound legs are included in the 2D and 3D illustrations in panels (a) and (b). Panels (c) and (d) display the southbound and northbound leg observations separately. Panels (e) and (f) show GLORIA water vapour (as in Fig. 9a and b), with data points characterized by ozone >65 ppbv and water vapour >12 ppmv marked by white diamonds (cf dashed black lines in panels (a), (c) and (d), colour-coded points in panel (b), and Gettelmann et al., 2011, their Sect. 4.3). Θ (K, solid and dashed grey lines, ΔΘ = 4 K) as derived from GLORIA's temperature and IFS horizontal wind $V_H$ (m s$^{-1}$, bold magenta lines) are superimposed in panels (e) and (f).

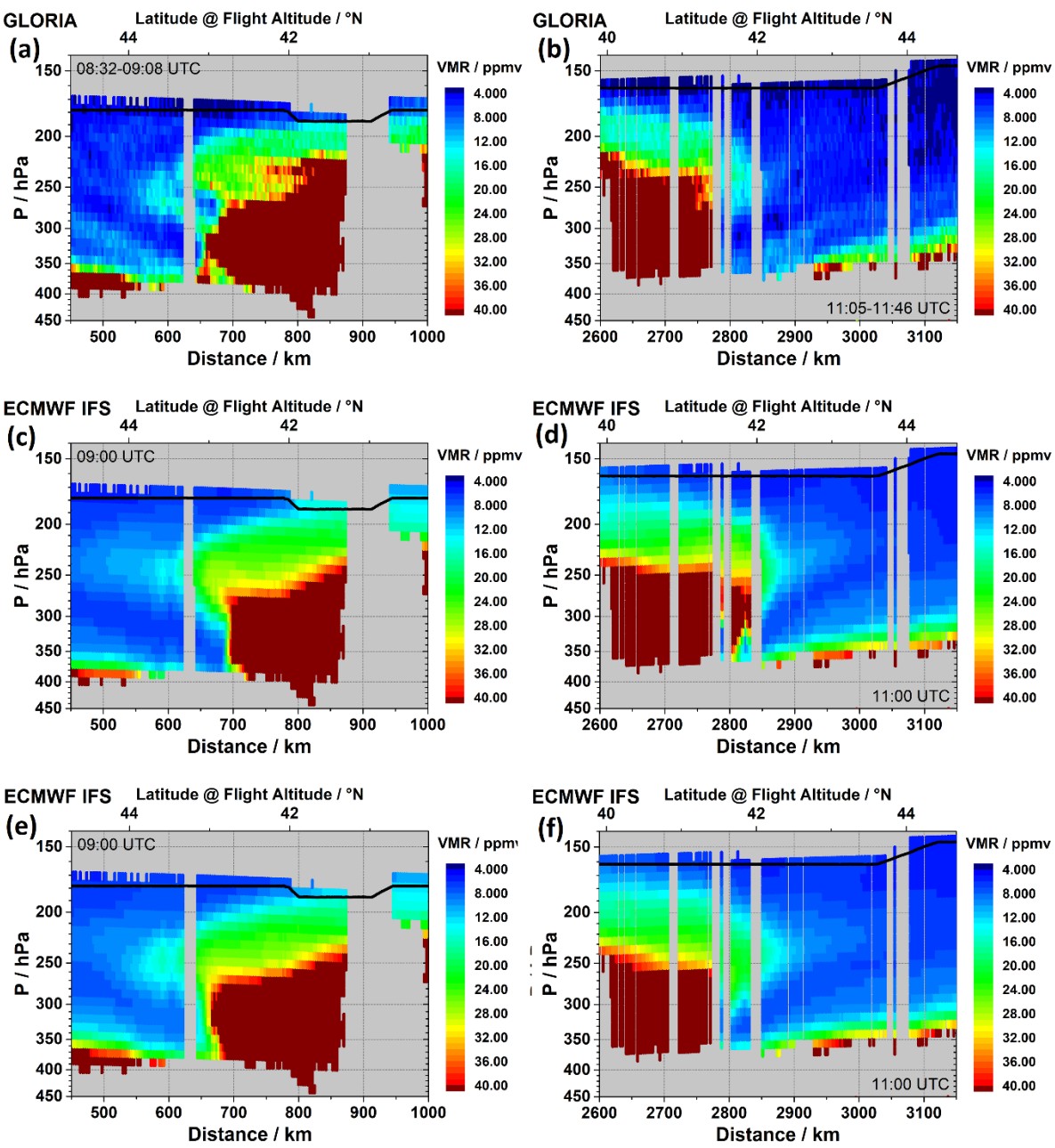

**Figure A1:** GLORIA water vapour (a, b) and IFS water vapour sampled at GLORIA tangent points (c,d) and using GLORIA observational filters (e,f). Black line in all panels: flight level.

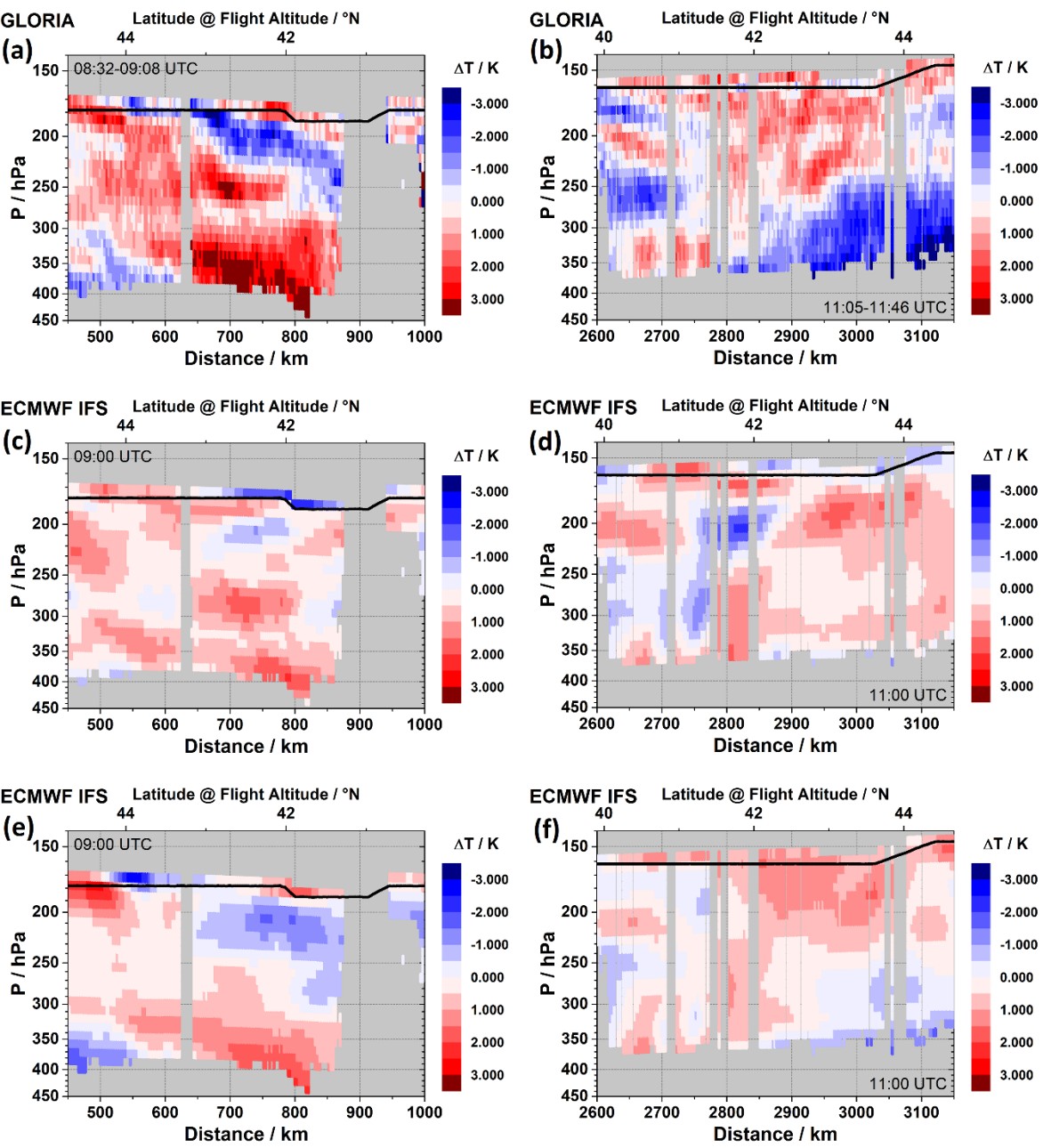

**Figure A2:** GLORIA residual temperature (a, b) and IFS residual temperature sampled at GLORIA tangent points (c,d) and using GLORIA observational filters (e,f). Black line in all panels: flight level.