# Peer review of "Mesoscale fine structure of a tropopause fold over mountains"

_Atmospheric Chemistry and Physics, 2018_

## Referee Comment (RC1) · Anonymous Referee #1 · 7 Aug 2018

The paper describes detailed observations of water vapor, ozone, and temperature at a tropopause fold in the vicinity of the polar frontal jet over Italy retrieved from measurements of GLORIA on HALOE . The data are compared to in situ measurements and high resolution model results. Observed fine structures and mixing between Stratosphere and Troposphere is discussed. The paper is well written and fits the scope of this journal well.

I have only some minor comments:

Page 3, line 25-26: actually already Weigel et al. (2012) (different to Ungermann et al. (2013), which is based on the same measurements) showed results of temperature and trace gas retrieval together at a topopause fold?

Page 6 or 7, Section 2.1: it should be mentioned, that the GLORIA data are cloud filtered and how they were filtered?

Page 16, line 29ff / Fig. 7d/f: do the "green" profiles really show dry stratospheric air or is there an issue with the measurement quality? They remain surprisingly close to the a priori values and a data gap is following in Fig. 7a? For Fig. 7c/d a logarithmic x-axis would probably be better?

Fig. 11a: I'm not sure if the 3d figure is really helpful here?The axis are difficult to read. Is it possible to improve it or, if not switch to a 2D Figure of H2O and O3 versus potential temperature?

Probably beyond the scope of this study, but it would be interesting, if the retrieval of additional trace gases is possible and how their distribution looks like?

---

## Referee Comment (RC2) · J. Gille (Referee) · 9 Aug 2018

Review of "Mesoscale fine structure of a tropopause fold over mountains"

General comments:

This is an excellent paper, presenting new observations with very high vertical and horizontal resolution of a tropopause fold over the Alps. The observations obtained by the GLORIA spectrometer on the HALO aircraft yielded temperature, water vapor and ozone data as a function of distance along the flight path. The observations were particularly good because GLORIA viewed perpendicular to the flight direction, and nearly along the jet stream and the fold, so that the effects of horizontal smoothing were small, and very clear cross-sections across the jet stream were obtained. These

were shown to agree with previous work on tropopause folds but provide much more detail because of the tracers of stratospheric and tropospheric air. The effects of mixing in and near the fold are clearly shown. The effects of gravity waves on mixing are also seen. They also showed that the ECMWF Integrated Forecast System (IFS) run at high resolution reproduced many of these features, but best agreement requires application of the GLORIA observational filter. The IFS did not reproduce some of the fine scale filaments observed, nor the gravity waves that appeared to facilitate mixing. The results are clearly and logically presented.

Specific scientific comments:

The title reflects the contents of the paper, and the abstract provides a complete summary. The methods are clearly spelled out, in clear and fluent English. Appropriate references are given to previous work. No parts of the paper should be eliminated. Some clarification is suggested in places.

Particular comments: p.1, l. 14; For those of us who do not immediately know which is the cyclonic or anticyclonic v side of the jet, at the initial mention, either here or in the text, please state which is north and south of the jet. This is done now, but should occur sooner. p. 5, l. 4; Explain what is meant by "linear limb observations". p.6, l. 11; how are clouds detected and their effects removed? l. 23; clarify- is temperature retrieved as a function of pressure, then put on an altitude scale using IFS data, or something else? l. 25; Please comment on the use of these frequencies for the temperature retrieval; they are considerably higher than often used p.7, l. 13; what is the vertical coordinate? p.9, l. 26ff; The plots in Figure 2 are very interesting, but not intuitively obvious. It would be useful if the authors could provide more of a physical explanation of what they are showing. Is it that if there is more water vapor in the layer above the tangent layer, the retrieval over-corrects, putting more water vapor in that layer, and less in the tangent layer? Is it the result of the perturbation being narrower than the vertical weighting function? Or something elses? p. 19, l. 27ff; How are the perturbations defined? Perturbed relative to what? p.21, ll 1 ff; The tracer-tracer plots are interesting

and appropriate Fig. 11a- difficult to make much out of the 3-D plot- is this primarily meant for people to look at online? l. 12: looks like H2O values < 310 K (red) are low, not enhanced; l. 24; looks like between 320 K and 340 K (green & blue), not higher l. 26; nterpretation would be aided greatly if the location of a contour of the core of the jet stream winds were shown, as in Figs. 10c,d.

p. 23, l.28; As noted above, the figures seem to show 310 to 340 K.

Minor technical points:

p.1, l. 23: "used" not needed. p.3, l. 7; diagnostics applied l.16; all vertical viewing angles p. 4, l.13; "Admittedly" not needed p.6, l. 23. "each" not needed l. 27ff; Is "data" a plural word? Some say so, in which case data are or were p.8, l.18; within the microwindows used p. 10, l. 10; originate p. 13, l.13/14. In the period considered p. 15 l. 23; a gray line p.18, l.18; Fig. 8b,d l. 20; alleviated is not the right word; do you mean reproduced? p. 23, l.28; do you mean identified carefully? Conscientiously not the right word.

---

## Author Comment (AC1) · 13 Sep 2018

**Author response to comments by Anonymous Referee #1: "Mesoscale fine structure of a tropopause fold over mountains"**

*Atmos. Chem. Phys. Discuss., https://doi.org/10.5194/acp-2018-625, in review, 2018*
W. Woiwode et al.

We thank Referee #1 for his/her time and valuable comments to improve the manuscript. In the following, we provide the original referee comments (italic letters), followed by our responses. Text added or modified in the revised manuscript is coloured in blue.

*The paper describes detailed observations of water vapor, ozone, and temperature at a tropopause fold in the vicinity of the polar frontal jet over Italy retrieved from measurements of GLORIA on HALOE. The data are compared to in situ measurements and high resolution model results. Observed fine structures and mixing between Stratosphere and Troposphere is discussed. The paper is well written and fits the scope of this journal well.*

We thank Referee #1 for this favourable statement.

**Minor comments:**

*Page 3, line 25-26: actually already Weigel et al. (2012) (different to Ungermann et al. (2013), which is based on the same measurements) showed results of temperature and trace gas retrieval together at a topopause fold?*

Here, we intended to address in situ observations only. We agree that Weigel et al. (2012) should be mentioned here. We rephrased P3/L25-26 as follows:

"…Simultaneous in situ measurements of both trace gases and temperature have not been applied up to now to characterize the mesoscale fine structure of tropopause folds. CRISTA-NF observations in July 2006 (Weigel et al., 2012) provided a first coarse perspective of a tropopause fold using this combination of parameters. Here, we present…"

*Page 6 or 7, Section 2.1: it should be mentioned, that the GLORIA data are cloud filtered and how they were filtered?*

We added at P6/L21: "…by Johansson et al. (2018). Prior to the retrieval, the binned spectra are cloud-filtered according to the cloud index method by Spang et al. (2004). A variable threshold value is applied, increasing linearly from 3.0 for the lowest limb-views to 1.8 at flight altitude. For the retrievals …" and provide the reference Spang et al. (2004) under "References":

Spang, R., Remedios, J. J., and Barkley, M. P.: Colour indices for the detection and differentiation of cloud types in infra-red limb emission spectra, Adv. Space Res., 33, 1041–1047, 2004.

To clarify that the binned spectra are cloud-filtered, we modified P6/L12: "… In post-flight data processing, spectra of the detector rows of each data cube are binned to reduce uncertainties. …"

*Page 16, line 29ff / Fig. 7d/f: do the "green" profiles really show dry stratospheric air or is there an issue with the measurement quality? They remain surprisingly close to the a priori values and a data gap is following in Fig. 7a? For Fig. 7c/d a logarithmic x-axis would probably be better?*

We confirm that there is no issue with the measurement quality. As recommended, we now adopted logarithmic x-axes for Figures 7c/d. The revised plots show that the water vapour profiles exhibit coherent fine structures different from the a priori even below 10 ppmv:

[Figure]

Fig. 11a: *I'm not sure if the 3d figure is really helpful here? The axis are difficult to read. Is it possible to improve it or, if not switch to a 2D Figure of $H_2O$ and $O_3$ versus potential temperature?*

Here, our intention was to illustrate how the observed mixing zone is situated in a $H_2O$-$O_3$-$\Theta$ space. For clarification, we now changed the perspective of the 3d figure and colour-coded only the data points associated with the mixing zone. We furthermore exchanged the order of Figures 11a and b. Considering also the comments by Referee #2, we updated Figure 11:

[revised manuscript text omitted]

*Probably beyond the scope of this study, but it would be interesting, if the retrieval of additional trace gases is possible and how their distribution looks like?*

Retrievals of more tracers are possible from the GLORIA observations (see Johansson et al., 2018 and supplement https://doi.org/10.5194/amt-11-4737-2018-supplement; further species are accessible). Here, we use $H_2O$ and $O_3$, since these species show strong gradients in the considered vertical range and are well accessible with GLORIA in terms of vertical resolution and uncertainties.

---

## Author Comment (AC2) · 13 Sep 2018

**Author response to comments by John Gille (Referee #2): "Mesoscale fine structure of a tropopause fold over mountains"**

*Atmos. Chem. Phys. Discuss., https://doi.org/10.5194/acp-2018-625, in review, 2018*
W. Woiwode et al.

We thank John Gille for his time and valuable comments to improve the manuscript. In the following, we provide the original referee comments (italic letters), followed by our responses. Text added or modified in the revised manuscript is coloured in blue.

*This is an excellent paper, presenting new observations with very high vertical and horizontal resolution of a tropopause fold over the Alps. The observations obtained by the GLORIA spectrometer on the HALO aircraft yielded temperature, water vapor and ozone data as a function of distance along the flight path. The observations were particularly good because GLORIA viewed perpendicular to the flight direction, and nearly along the jet stream and the fold, so that the effects of horizontal smoothing were small, and very clear cross-sections across the jet stream were obtained. These were shown to agree with previous work on tropopause folds but provide much more detail because of the tracers of stratospheric and tropospheric air. The effects of mixing in and near the fold are clearly shown. The effects of gravity waves on mixing are also seen. They also showed that the ECMWF Integrated Forecast System (IFS) run at high resolution reproduced many of these features, but best agreement requires application of the GLORIA observational filter. The IFS did not reproduce some of the fine scale filaments observed, nor the gravity waves that appeared to facilitate mixing. The results are clearly and logically presented.*

We thank the John Gille for this very positive statement.

**Specific scientific comments:**

*The title reflects the contents of the paper, and the abstract provides a complete summary. The methods are clearly spelled out, in clear and fluent English. Appropriate references are given to previous work. No parts of the paper should be eliminated. Some clarification is suggested in places.*

*Particular comments: p.1, l. 14; For those of us who do not immediately know which is the cyclonic or anticyclonic v side of the jet, at the initial mention, either here or in the text, please state which is north and south of the jet. This is done now, but should occur sooner.*

We modified P1/L13-15: "… The mesoscale fine structures of dry filaments at the cyclonic shear side north of the jet and tongues of moist air entraining tropospheric air into the stratosphere along the anticyclonic shear side south of the jet were clearly resolved by GLORIA observations. …"

*p. 5, l. 4; Explain what is meant by "linear limb observations".*

We added at P5/L4: "… linear limb observations (i.e. viewing perpendicular to the flight path, without azimuth panning) …"

*p.6, l. 11; how are clouds detected and their effects removed?*

We now added a short description of cloud filtering, please see reply to Referee #1.

*l. 23; clarify- is temperature retrieved as a function of pressure, then put on an altitude scale using IFS data, or something else?*

For clarification, we modified P6/L28: "All retrievals are based on geometric altitude levels. Associated pressures are interpolated from the HRES data."

*l. 25; Please comment on the use of these frequencies for the temperature retrieval; they are considerably higher than often used*

These spectral microwindows were successfully used in the past for retrievals of GLORIA data and the precursor instrument MIPAS-STR (see Johansson et al., 2018 and references therein). We added after P6/L24: "…These microwindows show a sufficient transparency at low altitudes. The used spectral transitions are suited well for a temperature retrieval, since they are sufficiently strong, clearly separable from other signatures, and characterized by different opacities and different temperature dependences. …"

*p.7, l. 13; what is the vertical coordinate?*

The vertical model coordinate are model levels (137), ranging from 0.01 hPa down to the surface (cf. https://www.ecmwf.int/en/forecasts/documentation-and-support/137-model-levels)

We added at P7/L14: "… 137 model levels …"

*p.9, l. 26ff; The plots in Figure 2 are very interesting, but not intuitively obvious. It would be useful if the authors could provide more of a physical explanation of what they are showing. Is that if there is more water vapor in the layer above the tangent layer, the retrieval over-corrects, putting more water vapor in that layer, and less in the tangent layer? Is it the result of the perturbation being narrower than the vertical weighting function? Or something elses?*

For clarification, we added at P9/L27: "…each other). The individual panels show how a single state element of the retrieval result (i.e. target parameter at altitude level indicated on the top of a panel) responds to an atmospheric grid point along viewing direction (characterized by geometric altitude and horizontal distance) in the true state (Ungermann, 2011; Ungermann et al., 2011). Thereby, "constructive" contributions (red) correspond with atmospheric grid points where a higher/lower value in the true state results in a higher/lower value in the retrieval result. On the other hand, "destructive" contributions (blue) correspond with atmospheric grid points where a higher/lower value in the true state, anti-intuitively, results in lower/higher value in the retrieval result. In this manner, the response of the retrieval and its weighting functions in the vertical and horizontal domain along the viewing direction are characterised.  In Figure 2 …"

We added the following reference for more detailed information:

Ungermann, J.: Tomographic reconstruction of atmospheric volumes from infrared limb-imager measurements, PhD thesis, Wuppertal University, 2011.

*p. 19, l. 27ff; How are the perturbations defined? Perturbed relative to what?*

For clarification, we added at P19/L27: "… perturbations (versus IFS background temperatures, see Sect. 2.2) …" (cf. P7/L25ff)

*p.21, ll 1 ff; The tracer-tracer plots are interesting and appropriate Fig. 11a- difficult to make much out of the 3-D plot- is this primarily meant for people to look at online?*

Here, our intention was to illustrate how the observed mixing zone is situated in $H_2O$-$O_3$-$\Theta$ space. For clarification, we revised the 3d correlation and colour-coded only the data points associated with the mixing zone (data points outside mixing region shown in grey). Furthermore, we shaded the regions outside the mixing a region in the 2d plots in grey for clarity.

[Figure]

**"Figure 11:** Tracer-tracer correlations of GLORIA's water vapour and ozone profiles during both tropopause fold passages as function of the observed potential temperatures. All observational data along south- and northbound legs are included in the 2D and 3D illustrations in panels (a) and (b). Panels (c) and (d) display the southbound and northbound leg observations separately. Panels (e) and (f) show GLORIA water vapour (as in Fig. 9a and b), with data points characterized by ozone >65 ppbv and water vapour >12 ppmv marked by white diamonds (cf dashed black lines in panels (a), **(c) and (d),** colour-coded points in panel (b), and Gettelmann et al., 2011, their Sect. 4.3). Θ (K, solid and dashed grey lines, ΔΘ = 4 K) as derived from GLORIA's temperature and IFS horizontal wind $V_H$ (m s⁻¹, bold magenta lines) are superimposed in panels (e) and (f)."

*l. 12: looks like H2O values < 310 K (red) are low, not enhanced;*

Here, we refer to slightly enhanced $H_2O$ values (up to ~50 ppmv max.) below 310 K found in panels (a)-(d). For clarification, we modified P21/L12-13: "…while the slightly enhanced $H_2O$-values (up to ~50 ppmv) below …"

*l. 24; looks like between 320 K and 340 K (green & blue), not higher*

Here, we refer to data points with ozone enhanced above 200 ppbv (blue and cyan). For clarification, we modified P21/L24-25: "Especially between 330 K and 350 K, we find enhanced water vapour values which are accompanied by notably enhanced ozone volume mixing ratios above 200 ppbv (Fig. 11c and d)."

*l.26; Interpretation would be aided greatly if the location of a contour of the core of the jet stream winds were shown, as in Figs. 10c,d.*

See new Figure 11 above: we superimposed contours of horizontal wind speed in the core region of the jet in panels (e) and (f).

*p. 23, l.28; As noted above, the figures seem to show 310 to 340 K.*

See above: at P21/L24-25 we referred to data points in the mixing region with ozone values above 200 ppbv. Here, we refer to the entire mixing region, i.e. all blue, cyan, green, yellow and red data points situated in the mixing region in the new Figure 11b (see above). Thereby, we want to highlight the change from a predominantly tropospheric correlation below 330 K (due to stratosphere-to-troposphere exchange at the cyclonic shear side and in the lower compartment of the fold) to a predominantly stratospheric correlation above (due troposphere-to-stratosphere exchange at the anticyclonic shear side). For clarification, we rephrased P23/L28 as follows: "…Mixing takes place in the interval 300 K < Θ < 350 K, with stratosphere-to-troposphere exchange taking place predominantly below 330 K and troposphere-to-stratosphere exchange above. Two major …"

For clearer identification of the locations of mixing regions in Figures 11e and 11f relative to the jet stream and the isentropes we further revised these panels. All data points attributed to the ExTL mixing zone are now marked by white diamonds for easier reading. For a better assignment of potential temperature levels, we now superimposed GLORIA Θ contours.

For a better assignment of the mixing regions in the tracer correlations and vertical cross sections, we further revised the discussion from P21/L9 to P22/L9 (see reply to Referee #1).

**Technical points:**

*p.1, l. 23: "used" not needed.*

Here, our intention is to indicate that resolutions other than 9 km are also possible with the IFS.

*p.3, l. 7; diagnostics applied*

done

*l.16; all vertical viewing angles*

done

*p. 4, l.13; "Admittedly" not needed*

If acceptable, we would like to stay with this wording

*p.6, l. 23. "each" not needed*

Here we wanted to express that 2 spectral microwindows including each 2 spectral lines were used. For clarification, we rephrased P6/L23 as follows: "… using two times two rotational-vibrational transitions… "

*l. 27ff; Is "data" a plural word? Some say so, in which case data are or were*

We rephrased: "… The data are available every six hours and are hereafter …"

*p.8, l.18; within the microwindows used*

done

*p. 10, l. 10; originate*

done

*p. 13, l.13/14. In the period considered*

done

*p.15 l. 23; a gray line*

done

*p.18, l.18; Fig. 8b,d*

done

*l. 20; alleviated is not the right word; do you mean reproduced?*

done

*p. 23, l.28; do you mean identified carefully? Conscientiously not the right word.*

We removed "conscientiously"